# Local cortical desynchronization and pupil-linked arousal differentially shape brain states for optimal sensory performance

**Leonhard Waschke\*, Sarah Tune, Jonas Obleser\***

Department of Psychology, University of Lübeck, Lübeck, Germany

**Abstract** Instantaneous brain states have consequences for our sensation, perception, and behaviour. Fluctuations in arousal and neural desynchronization likely pose perceptually relevant states. However, their relationship and their relative impact on perception is unclear. We here show that, at the single-trial level in humans, local desynchronization in sensory cortex (expressed as time-series entropy) versus pupil-linked arousal differentially impact perceptual processing. While we recorded electroencephalography (EEG) and pupillometry data, stimuli of a demanding auditory discrimination task were presented into states of high or low desynchronization of auditory cortex via a real-time closed-loop setup. Desynchronization and arousal distinctly influenced stimulus-evoked activity and shaped behaviour displaying an inverted u-shaped relationship: States of intermediate desynchronization elicited minimal response bias and fastest responses, while states of intermediate arousal gave rise to highest response sensitivity. Our results speak to a model in which independent states of local desynchronization and global arousal jointly optimise sensory processing and performance.

**\*For correspondence:**
leonhard.waschke@uni-luebeck.de
(LW);
jonas.obleser@uni-luebeck.de (JO)

**Competing interests:** The authors declare that no competing interests exist.

## Introduction

The way we sense and perceive our environment is not determined by physical input through the senses alone. The dynamics of ongoing brain activity affect the build-up of sensory representations and our conscious perception of the physical world. Recently, instantaneous fluctuations of both pupil-linked arousal (*McGinley et al., 2015b*; *Lee et al., 2018*; *Pfeffer et al., 2018*) and neural desynchronization (*Curto et al., 2009*; *Marguet and Harris, 2011*; *Pachitariu et al., 2015*) have been highlighted as sources of such sensory and perceptual variation: Arousal and cortical desynchronization are two ways of characterizing the brain state, which strongly influences sensory cortical responses, the encoding of information, thus perception and ultimately behaviour.

The term arousal here and henceforth is used to refer to the general level of alertness which likely traces back to neuromodulatory activity and is associated with the ascending reticular activating system (ARAS). Pupil-linked arousal, which captures locus coeruleus-norepinephrine activity (LC–NE; *Aston-Jones and Cohen, 2005*; *Joshi et al., 2016*; *Reimer et al., 2016*) has been shown to influence sensory evoked activity (*McGinley et al., 2015a*; *McGinley et al., 2015b*; *Gelbard-Sagiv et al., 2018*) and the processing of task-relevant information (*Murphy et al., 2014*; *Lee et al., 2018*). Despite evidence for an inverted u-shaped relation of tonic LC–NE activity to performance long suspected from the Yerkes-Dodson law (*Yerkes and Dodson, 1908*), the precise associations between arousal, sensory processing, and behaviour are underspecified: Although optimal performance at intermediate levels of arousal has reliably been observed (*Murphy et al., 2014*; *McGinley et al., 2015b*; *McGinley et al., 2015a*; *van den Brink et al., 2016*; *Faller et al., 2019*),

reports of linear effects on performance (*Gelbard-Sagiv et al., 2018*) or evoked activity (*Neske and McCormick, 2018*) in different tasks and species complicate this picture.

In a separate line of experimental work in non-human animals, relatively high neural desynchronization yielded improved encoding and representation of visual (*Goard and Dan, 2009*; *Pinto et al., 2013*; *Beaman et al., 2017*) as well as auditory input (*Marguet and Harris, 2011*; *Pachitariu et al., 2015*; *Sakata, 2016*). Such periods of desynchronization are characterized by reduced noise correlations in population activity, and these patterns are commonly referred to as desynchronized cortical states. They likely result from subtle changes in the balance of excitatory and inhibitory activity (*Renart et al., 2010*; *Haider et al., 2013*). Notably, behaviourally relevant changes in cortical desynchronization have been suggested to trace back to attention-related changes in thalamo-cortical interactions (*Harris and Thiele, 2011*). Thus, such desynchronization states can be expected to be of local nature and be limited to sensory cortical areas of the currently attended sensory domain (*Beaman et al., 2017*). Although local desynchronization and perceptual performance are positively linked in general (*Beaman et al., 2017*; *Speed et al., 2019*), the exact shape of their relationship (e. g., linear vs. quadratic) is unclear. Most notably, evidence for a similar mechanism in humans has remained elusive.

On the one hand, a tight link of pupil size and desynchronization has been claimed (*McCormick, 1989*; *McCormick et al., 1991*; *McGinley et al., 2015a*; *Vinck et al., 2015*). On the other hand, both measures have also been found to be locally unrelated (*Beaman et al., 2017*; *Okun et al., 2019*). As of now, pupil-linked arousal and local cortical desynchronization may or may not be distinct signatures of the same underlying process: Varying noradrenergic and cholinergic activity could influence both, local cortical activity and the more global measure of pupil size via afferent projections from brain-stem nuclei (*Harris and Thiele, 2011*). In sum, it is, first, unclear how pupil-linked arousal and local cortical desynchronization precisely shape sensory processing and perceptual performance in humans. Second, the interrelation of both measures and their potentially shared underlying formative process lacks specification.

Here, we set out to test the relationship of local desynchronization states and pupil-linked arousal, and to specify their relative impact on sensory processing and perception in healthy human participants. We recorded EEG and pupillometry while participants performed a challenging auditory discrimination task. We modelled ongoing neural activity, sensory processing, and perceptual performance based on both local cortical desynchronization and pupil-linked arousal. This way we were able to test the interrelations of both measures but also to directly inspect their shared as well as exclusive influence on sensory processing and behaviour. Specifically, the effects of local cortical desynchronization and pupil-linked arousal on perceptual sensitivity as well as response criterion were analysed.

A closed-loop real-time algorithm calculated on-line an information theoretic proxy of auditory cortical desynchronization (weighted permutation entropy, WPE; *Fadlallah et al., 2013*; *Waschke et al., 2017*) based on EEG signal arising predominantly from auditory cortices. Of note, WPE as a proxy of desynchronization is tailored to the analysis of electrophysiological time series: It captures oscillatory as well as non-oscillatory contributions as a time-resolved estimate of desynchronization (see Materials and methods for details). Importantly, EEG entropy calculated for a previously published data set (*Sarasso et al., 2015*) aptly tracks changes in excitatory and inhibitory (E/I) cortical activity that occur under different anaesthetics (*Figure 2—figure supplement 1*). Also, EEG entropy as measured in the present data aligns closely with the spectral exponent, a previously suggested measure of E/I (*Figure 2—figure supplement 1*; *Gao et al., 2017*; *Waschke et al., 2017*). Entropy of EEG signals thus is not only sensitive to the basic features of desynchronization (e.g. reduced oscillatory power) but also captures changes in a central underlying mechanism (E/I balance).

We used this measure of ongoing desynchronization to trigger stimulus presentation during relatively synchronized and desynchronized states, respectively. A continuously adapting criterion enabled us to effectively sample the whole desynchronization state space (*Jazayeri and Afraz, 2017*). Such a closed-loop set up allows for selective stimulation during specific states of brain activity while accounting for changes in the appearance of those states and hence represents a powerful tool with a multitude of potential applications in research but also therapy (*Sitaram et al., 2017*; *Ezzyat et al., 2018*). To evaluate the interrelation of pre-stimulus desynchronization with simultaneously acquired pupil-linked arousal as well as their influence on stimulus-related activity we

employed linear mixed-effect models. Furthermore, psychophysical models were used to evaluate the impact of desynchronization and arousal on perceptual sensitivity, response criterion, and response speed.

Although local cortical desynchronization and pupil-linked arousal were weakly positively correlated, both did not only shape the ongoing EEG activity into distinct states, but also differentially influenced sensory processing at the level of single trials: On the one hand, phase-locked activity in low frequencies as well as stimulus-related gamma power over auditory cortices was highest following intermediate levels of pre-stimulus desynchronization. On the other hand, low-frequency power during and after a stimulus increased linearly with pre-stimulus arousal. Response criterion and speed exhibited an inverted u-shaped relationship with local cortical desynchronization, where intermediate desynchronization corresponded to minimal response bias and fastest responses. An analogous relationship was found for arousal and sensitivity, revealing highest sensitivity at intermediate arousal levels.

Our results speak to a model in which global arousal states and local desynchronization states jointly influence sensory processing and performance. While fluctuations in arousal are likely realized by afferent cholinergic and noradrenergic projections into sensory cortical areas (*Robbins, 1997*; *Carter et al., 2010*), desynchronization states might result from efferent feedback connections (*Harris and Thiele, 2011*; *Zagha et al., 2013*).

## Results

We recorded EEG and pupillometry while participants (N = 25; 19–31 years old) performed an auditory pitch discrimination task. On each trial participants were presented with one tone, taken from a set of seven pure tones (increasing pitch from tone 1 through tone 7), and had to decide whether that tone was rather high or low in pitch with regard to the overall set of tones. Participants thus compared each tone to an implicit standard, the median (=mean) pitch of the set. This yielded in all participants a valid psychometric function mapping stimulus pitch to perceptual decisions (see *Figure 5—figure supplement 2*).

Critically, by means of a real-time closed-loop algorithm (see *Figure 1*), tones were presented during states of relatively high or low entropy of auditory cortical EEG, a proxy of local cortical desynchronization. By collapsing offline across the whole experiment, we obtained data that covered the whole range of desynchronization states occurring in a given participant (*Jazayeri and Afraz, 2017*). We then combined (generalized) linear mixed-effects models and psychophysical modelling to test the effects of local cortical desynchronization as well as pupil-linked arousal on (1) ongoing as well as sensory-related EEG activity, and on (2) perceptual performance.

### Real-time closed-loop algorithm dissociates desynchronization states

Entropy of EEG signals emerging from auditory cortices was calculated with the help of an established, functional–localizer-based spatial filter (see *Figure 2a*; *de Cheveigné and Simon, 2008*; *Herrmann et al., 2018*) and a custom real-time algorithm (*Figure 1*). Source projection of localizer data which were used to construct the subject-specific spatial filters revealed predominantly auditory cortical regions as generators (*Figure 2a*).

Note that the distribution of entropy values which provided the basis for the classification of relatively high vs. relatively low desynchronization states was updated continuously, with two crucial consequences: First, this approach minimized the potential impact of slow drifts in desynchronization on brain state classification. Second, the continuously updated criterion allowed us to, effectively, sample the whole state space of local desynchronization states: Depending on the current distribution, the same absolute entropy value could be classified as a high state, for example in the beginning of the experiment, and as a low state half an hour later. This focus on local, short-lived states resulted in widely overlapping pre-stimulus entropy distributions of high and low states (*Figure 2c*) which were then used as continuous predictor alongside the equally continuous pupil-size in all subsequent analyses.

Demonstrating the performance of the real-time algorithm, average entropy time-courses were elevated for all classified-high compared to all classified-low states in a 200 ms pre-stimulus window (all p<0.001, FDR corrected; *Figure 2b*). Note that this result is non-trivial. Since we continuously

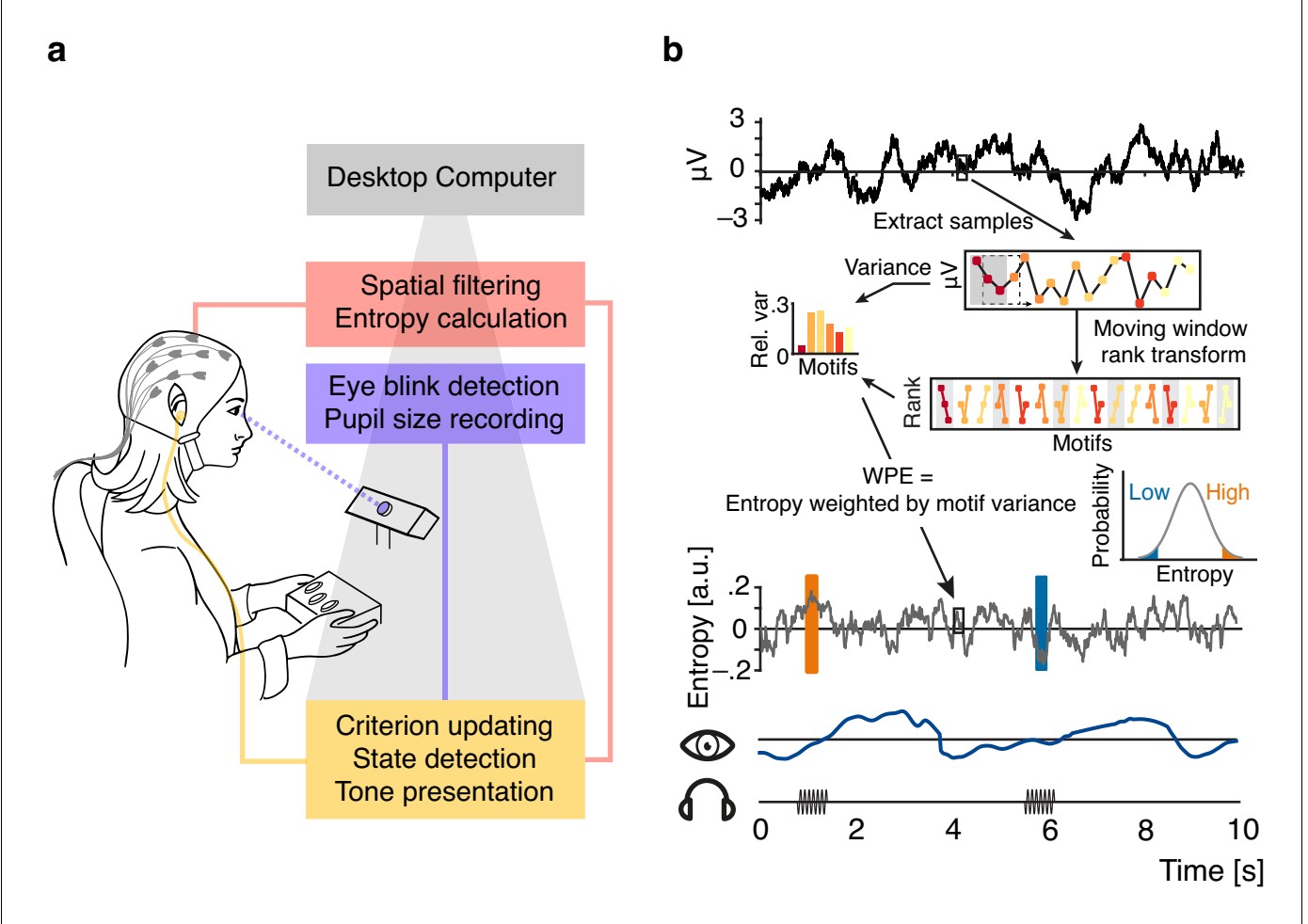

**Figure 1.** Illustration of the real-time closed-loop setup to track states of desynchronization. (a) Setup: EEG signal was spatially filtered before entropy calculation. Pupil size was recorded and monitored consistently. Pure tone stimuli were presented via in-ear headphones during states of high or low entropy of the incoming EEG signal. (b) Schematic representation of the real-time algorithm: spatially filtered EEG signal (one virtual channel) was loaded before entropy was calculated using a moving window approach (illustrated for 18 samples in the upper box; 200 samples were used in the real-time algorithm). Voltage values were transformed into rank sequences ('motifs') separated by one sample (lower box; *Equation 1* in Materials and methods; different colours denote different motifs), and motif occurrence frequencies were weighted by the variance of the original EEG data constituting each occurrence (*Equations 3 and 4*). Each entropy value was calculated based on the resulting conditional probabilities of 200 samples, before the window was moved 10 samples forward (i.e., effectively down-sampling to 100 Hz). Inset: The resulting entropy time-course was used to build a continuously updated distribution (forgetting window = 30 s). Ten consecutive entropy samples higher than 90% (or lower than 10%) of the currently considered distribution of samples defined states of relatively high and low desynchronization, respectively. Additionally, pupil size was sampled continuously.

updated the criterion for state detection, in theory, states classified online as high and low could have yielded the same average entropy across the entire experiment.

In contrast, pupil diameter time-courses did not differ between high and low entropy states at any point in time (all p>0.1) nor did the distributions of pre-stimulus pupil diameters (*Figure 2c*). In line with previous research (*Reimer et al., 2014*), pupil size and entropy in the pre-stimulus time window were positively related ($\beta$ = 0.02, SE = 0.01, p=0.02). Pupil size explained less than 1% of the variance in EEG entropy.

Furthermore, auditory cortical desynchronization and pupil linked arousal, as approximated by EEG entropy and pupil size, displayed different autocorrelation functions (*Figure 2b*). While EEG entropy states were self-similar on an approximate ~500 ms scale, states of pupil size extended over several seconds.

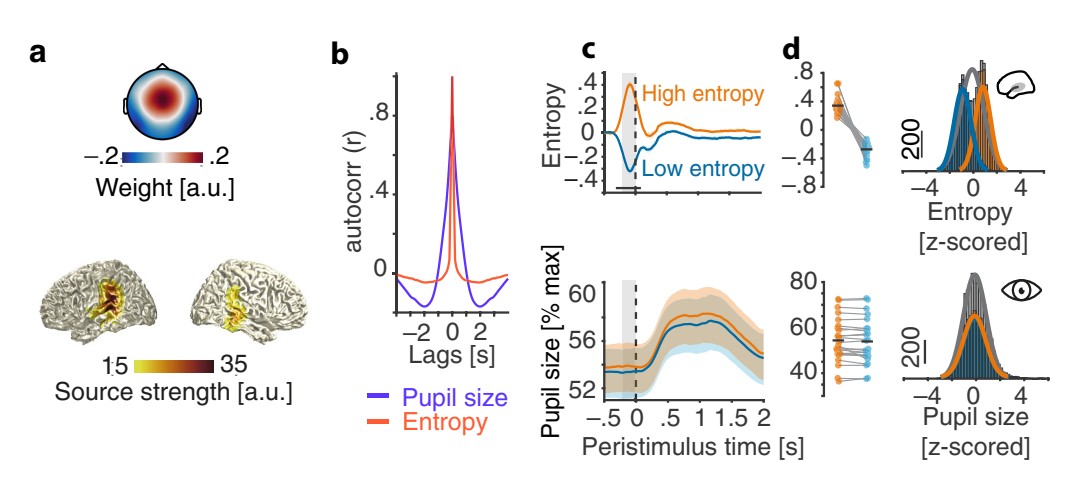

**Figure 2.** Evaluation of the real-time closed-loop setup for states of local desynchronization and arousal. (**a**) Grand average spatial filter weights based on data from an auditory localizer task (top) and grand average source projection of the same data (masked at 70% of maximum; bottom). (**b**) Autocorrelation functions for EEG entropy (red) and pupil size time courses (blue). Entropy states are most self-similar at ~500 ms (~2 Hz) and pupil states at ~2 s (~0.5 Hz). (**c**) Grand average time-courses of entropy (upper panel) and pupil diameter (lower panel) for low-entropy (blue) and high-entropy states (orange) ± standard error of the mean (SEM). Subject-wise averages in the pre-stimulus time-window (−200–0 ms, grey boxes) in right panels. Entropy was logit transformed and baseline corrected to the average of the preceding 3 s for illustration. Pupil size was expressed as percentage of each participant's maximum pupil diameter across all pre-stimulus time-windows. (**d**) Histograms and fitted distributions of absolute z-scored pre-stimulus entropy (top) and z-scored pupil size (bottom) for low-entropy states (blue), high-entropy states (orange), and both states combined (grey). Note the independence of entropy states and pupil states.

The online version of this article includes the following figure supplement(s) for figure 2:

**Figure supplement 1.** EEG entropy as a marker of E/I balance based on anaesthesia recordings from *Sarasso et al. (2015)*.

Most relevant to all further analyses, we conclude that states of local cortical desynchronization in auditory cortex and pupil-linked arousal predominantly occurred independently of each other.

## Local cortical desynchronization and pupil-linked arousal pose distinct states of ongoing activity

To dissociate the corollaries of local cortical desynchronization and pupil-linked arousal on ongoing EEG activity, we modelled single trial pre-stimulus oscillatory power over auditory cortical areas as a function of pre-stimulus entropy and pupil diameter by jointly including them as predictors in linear mixed-effects models. Of note, non-baselined values of EEG entropy and pupil size were used as predictors but baseline values of EEG entropy were included as covariates to control for the influence of slow temporal drifts. This approach has been suggested previously (*Senn, 2006*), is widely used in functional imaging (*Kay et al., 2008*), and is more reliable than conventional baseline subtraction methods (*Alday, 2019*). All analyses of ongoing or stimulus-related EEG activity were carried out on the spatially filtered EEG signal, allowing us to concentrate on brain activity dominated by auditory cortical regions.

As expected based on the definition of entropy and earlier results (*Waschke et al., 2017*), these analyses revealed a negative relationship of entropy and oscillatory power within the pre-stimulus time window (−200–0 ms; *Figure 3*). With increasing pre-stimulus entropy, low-frequency pre-stimulus power decreased (1–8 Hz, linear: β = −0.18, SE = 0.01, p<0.001; quadratic: β = 0.03, SE = 0.009, p<.005; *Supplementary file 1*). Gamma power (40–70 Hz) also decreased (linear: β = −0.18, SE = 0.01, p<0.001; *Supplementary file 4*). Gamma power was lowest at intermediate entropy levels (quadratic effect; β = 0.06, SE = 0.009, p<0.001). Furthermore, EEG entropy was negatively related to pre-stimulus alpha power (8–12 Hz, β = −0.29, SE = 0.01, p<0.001; *Figure 3—figure supplement 1* & *Supplementary file 2*) and beta power (14–30 Hz, β = −0.32, SE = 0.01, p<0.001, *Figure 3—figure supplement 1* & *Supplementary file 3*). Auditory EEG entropy hence aptly approximates the degree of auditory cortical desynchronization over a wide range of frequencies.

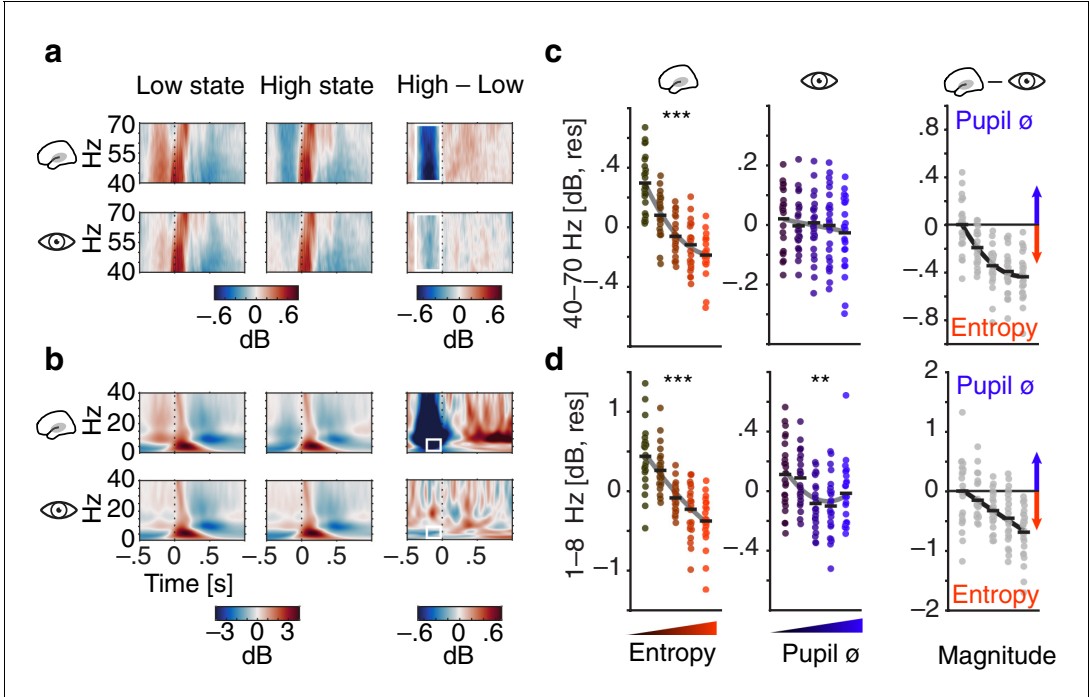

**Figure 3.** Contribution of pre-stimulus entropy and pupil size to ongoing auditory cortical EEG activity. (a) Grand average gamma power across time (40–70 Hz, baselined to the whole trial average, in dB) for low states (left), high states (middle) and the difference of both (right). Entropy states are shown in the upper panel, pupil states in the lower panel. Dashed line represents tone onset, white rectangle outlines the pre-stimulus window of interest. (b) As in (a) but for 0–40 Hz. (c) Mean-centred single subject (dots) and grand average gamma power (black lines) in the pre-stimulus time-window (−0.4–0 s), residualized for baseline entropy and pupil size, shown for five bins of increasing pre-stimulus entropy (left) and pupil size (residualized for entropy baseline and pre-stimulus entropy, middle). Grey line represents average fit, red colours show increasing entropy, blue colours increasing pupil size. Effects of entropy and pupil size are contrasted in the right panel. (d) As in (c) but for low-frequency power (1–8 Hz). Note the different y-axis range between entropy and pupil effects. All binning for illustrational purposes only. ***p<0.0001, **p<0.001.

The online version of this article includes the following figure supplement(s) for figure 3:

**Figure supplement 1.** Ongoing activity in the alpha and beta band as a function of EEG entropy and pupil size.

Analogously, pupil size was associated with a decrease in pre-stimulus low-frequency power (1–8 Hz, linear: $\beta = -0.04$, SE = 0.01, p<0.001; quadratic: $\beta = 0.016$, SE = 0.006, p<0.05; *Supplementary file 1*) but did not display a substantial relationship with gamma power (all p>0.2 see *Figure 3*; *Supplementary file 4*). Notably, pupil size was positively related with pre-stimulus beta power (14–30 Hz, $\beta = 0.04$, SE = 0.01, p<0.001; *Figure 3—figure supplement 1* and *Supplementary file 3*) but not with alpha power (all p>0.3).

To directly compare the relative contribution of EEG entropy and pupil size on ongoing EEG activity, respectively, we computed a Wald statistic ($Z_{Wald}$). The Wald statistic puts the difference between two estimates from the same model in relation to the standard error of their difference. The resulting Z-value can be used to test against equality of the two estimates. The stronger negative linear link of EEG entropy with low-frequency power compared to pupil size was supported by the Wald test ($Z_{Wald} = 9.1$, p<0.001). Put differently, in these stimulus-free periods in auditory cortex, low-frequency power was low given strong desynchronization, while it was additionally, yet more weakly, influenced by pupil-linked arousal. Notably, both patterns of results did not hinge on the exact choice of frequency ranges.

High-desynchronization states were thus characterized by reduced oscillatory broad-band power overall, while high-arousal states were accompanied by a decrease in low-frequency power and an increase in higher-frequency (beta) power.

# Differential effects of local desynchronization and pupil-linked arousal on auditory evoked activity

Next, to investigate the influence of those pre-stimulus states on sensory processing, we tested the impact of local cortical desynchronization and pupil-linked arousal in this pre-stimulus time window on auditory, stimulus-evoked EEG activity. Analogous to the procedure outlined above, we used linear mixed-effects models to estimate the effects of entropy and pupil size on sensory evoked power and phase coherence over auditory cortices. Note that we modelled continuous variables instead of an artificial division into high vs. low states. While low-frequency phase coherence quantifies how precise in time neural responses appear across trials, low-frequency power captures the magnitude of neural responses regardless of their polarity (*Tallon-Baudry et al., 1996*; *Makeig et al., 2004*). In addition, high-frequency power after stimulus onset likely originates from sensory regions and depicts sensory processing (*Tiitinen et al., 1993*). If EEG entropy and pupil size entail perceptual relevance, they should also influence sensory processing as approximated by the outlined measures. Please note that all measures of sensory processing were based on artefact-free EEG data.

First, we found low-frequency single-trial phase coherence after stimulus onset, a measure quantifying the consistency of phase-locked responses on a trial-wise basis (see Materials and methods for details), to increase with pre-stimulus entropy (1–8 Hz, 0–400 ms; $\beta = 0.05$, SE = 0.01, p<0.001, *Figure 4a,d*). Additionally, phase coherence did not only increase with pre-stimulus entropy but saturated at intermediate levels, as evidenced by a negative quadratic effect ($\beta = -0.02$ SE=0.009, p=0.02, *Supplementary file 9*).

Of note, there was no comparable relationship of pupil size and single-trial phase coherence (1–jITC, see Materials and methods for details; $\beta = -0.005$, SE = 0.006, p=0.5; $Z_{Wald} = 1.5$, p=0.1). Phase-locked responses hence increased with pre-stimulus auditory cortical desynchronization but were unaffected by variations in arousal.

Second, we observed a linear decrease of low-frequency power after stimulus onset, as a function of pre-stimulus entropy (1–8 Hz, 0–400 ms; $\beta = -0.02$, SE = 0.01, p=0.017, *Figure 4b,e*). In contrast, pre-stimulus pupil size did not affect post-stimulus low-frequency power significantly ($\beta = 0.015$, SE = 0.011, p=0.2; *Supplementary file 5*). Visual inspection of *Figure 4* yields increased post-stimulus desynchronization that occurs after the evoked response as the likely source of the EEG entropy related decrease in stimulus-evoked low-frequency power. Therefore, stimulus-induced activity in low frequencies changed linearly with auditory cortical desynchronization but remained unaltered under changing levels of pupil-linked arousal ($Z_{Wald} = 2.6$, p=0.009). Notably, post-stimulus oscillatory power in the alpha band increased linearly with pupil linked arousal ($\beta = 0.033$, SE = 0.01, p<.005; *Figure 4—figure supplement 2* & *Supplementary file 6*) but not with auditory cortical desynchronization ($\beta = -0.008$, SE = 0.009, p=0.5). Oscillatory power in the beta band was neither substantially linked to pre-stimulus auditory cortical desynchronization nor pupil-linked arousal (all p>0.2, see *Supplementary file 7*).

Third, we detected linearly increasing post-stimulus gamma power, representing early auditory evoked activity, with rising pre-stimulus entropy (40–70 Hz, 0–400 ms; $\beta = 0.04$, SE = 0.01, p<0.001, *Figure 4c,f*). Conversely, post-stimulus gamma power showed a tendency to decrease with growing pre-stimulus pupil size that did not reach statistical significance ($\beta = -0.016$, SE = 0.01, p=0.1; *Supplementary file 8*). Auditory evoked gamma power hence was inversely influenced by two different measures of brain state: while it increased with local cortical desynchronization, it decreased with growing arousal ($Z_{Wald} = 3.6$, p=0.0003). Notably, neither local desynchronization nor pupil size had any effect on the tone-evoked activity when expressed as event-related potentials (see *Figure 4—figure supplement 1*).

Overall, single-trial auditory sensory evoked activity was differentially influenced by desynchronization and arousal. While only higher local desynchronization was associated with increased phase-locked responses, only arousal was positively linked to stimulus-induced activity. In addition, with local desynchronization showing a positive and arousal a negative link to stimulus-evoked gamma power, both measures exert opposite influences on the early processing of auditory information.

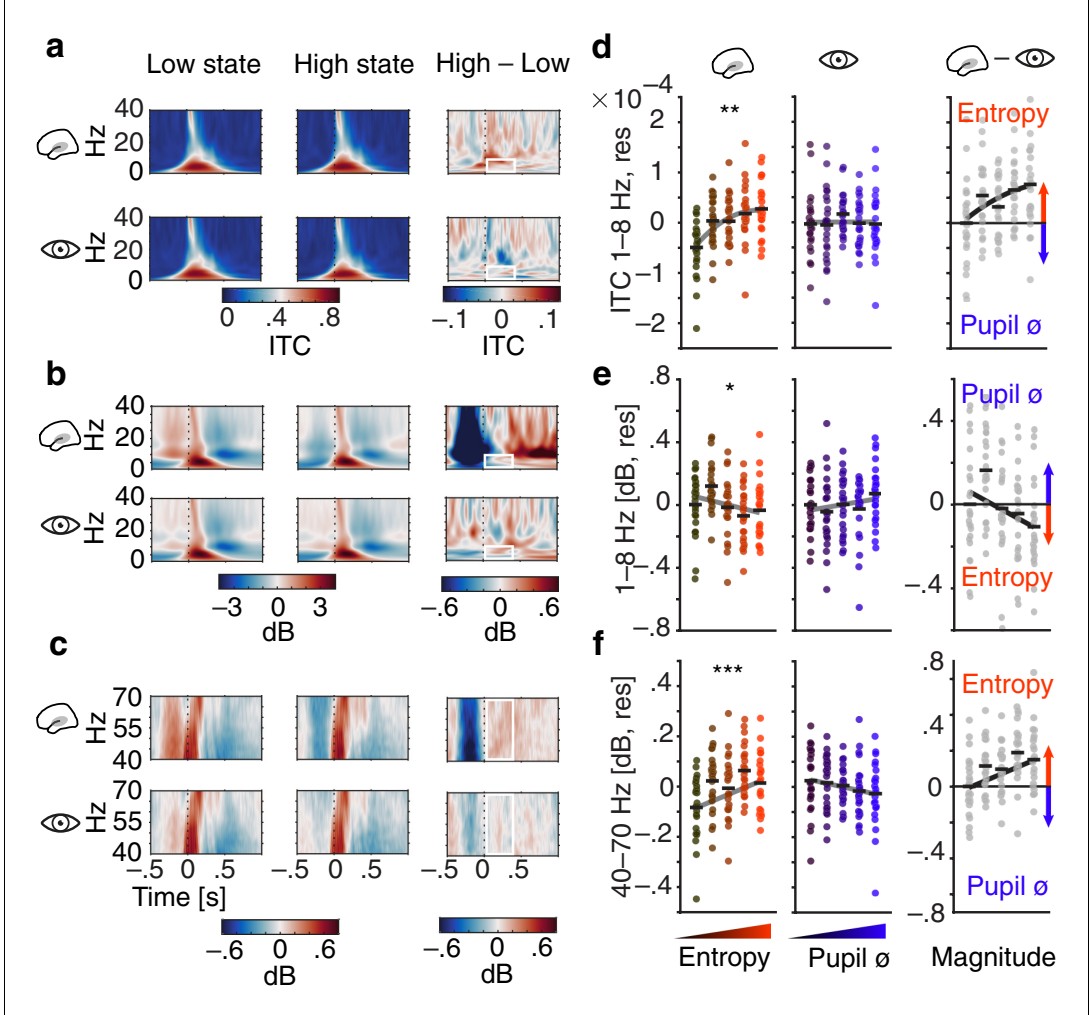

**Figure 4.** Influence of pre-stimulus entropy and pupil size on tone-related activity. (a) Grand average ITC (0–40 Hz) across time for low states (left), high states (middle) and the difference of both (right). Entropy states shown in the upper, pupil states in the lower panel. Dashed black lines indicate tone onset, white rectangles the post-stimulus window of interest. (b) As in (a) but for low-frequency power (0–40 Hz, baselined the average of the whole trial). (c) As in (b) but for gamma power (40–70 Hz). (d) Mean centred single subject (dots) and grand average ITC (black lines), residualized for baseline entropy and pupil size, in the post-stimulus time-window (0–.4 s, 1–8 Hz) for five bins of increasing pre-stimulus entropy (left) and pupil size (residualized for entropy baseline and pre-stimulus entropy, middle). Grey line represents average fit, red colours increasing entropy, blue colours increasing pupil size. Effects of entropy and pupil size are contrasted in the right panel. (e) As in (d) but for post-stimulus low-frequency power (0–.4 s, 1–8 Hz). (f) As in (e) but for post-stimulus gamma power (0–.4 s, 40–70 Hz). Again, all binning for illustrational purposes only. ***$p < 0.0001$, **$p < 0.001$, *$p < 0.05$.

The online version of this article includes the following figure supplement(s) for figure 4:

**Figure supplement 1.** Grand average ERPs for increasing pre-stimulus entropy and pupil size.

**Figure supplement 2.** Tone-related activity in the alpha and beta band as a function of pre-stimulus EEG entropy and pupil size.

## Local desynchronization and arousal differently impact perceptual performance

To examine the impact of desynchronization and arousal on perceptual performance, we modelled binary response behaviour ('high' vs. 'low') as a function of stimulus pitch, pre-stimulus local desynchronization, and arousal using generalized linear mixed-effects models (see *Statistical analyses* for details). In brief, this statistical approach describes binary choice behaviour across the set of used tones and thus also yields a psychometric function, but the generalized linear framework allows us to include the neural predictors of interest. Two parameters of the resulting functions were of interest to the current study: (1) the threshold of the psychometric function represents the response criterion; (2) the slope of the psychometric function expresses perceptual sensitivity. Additionally, we

tested the influence of local desynchronization and arousal on response speed (i.e., the inverse of response time, in s$^{-1}$). Note that models always included linear as well as quadratic terms in order to test the shape of the investigated brain-behaviour relationships.

Participants were least biased and answered fastest at intermediate levels of pre-stimulus desynchronization: pre-stimulus entropy displayed a negative quadratic relationship with response

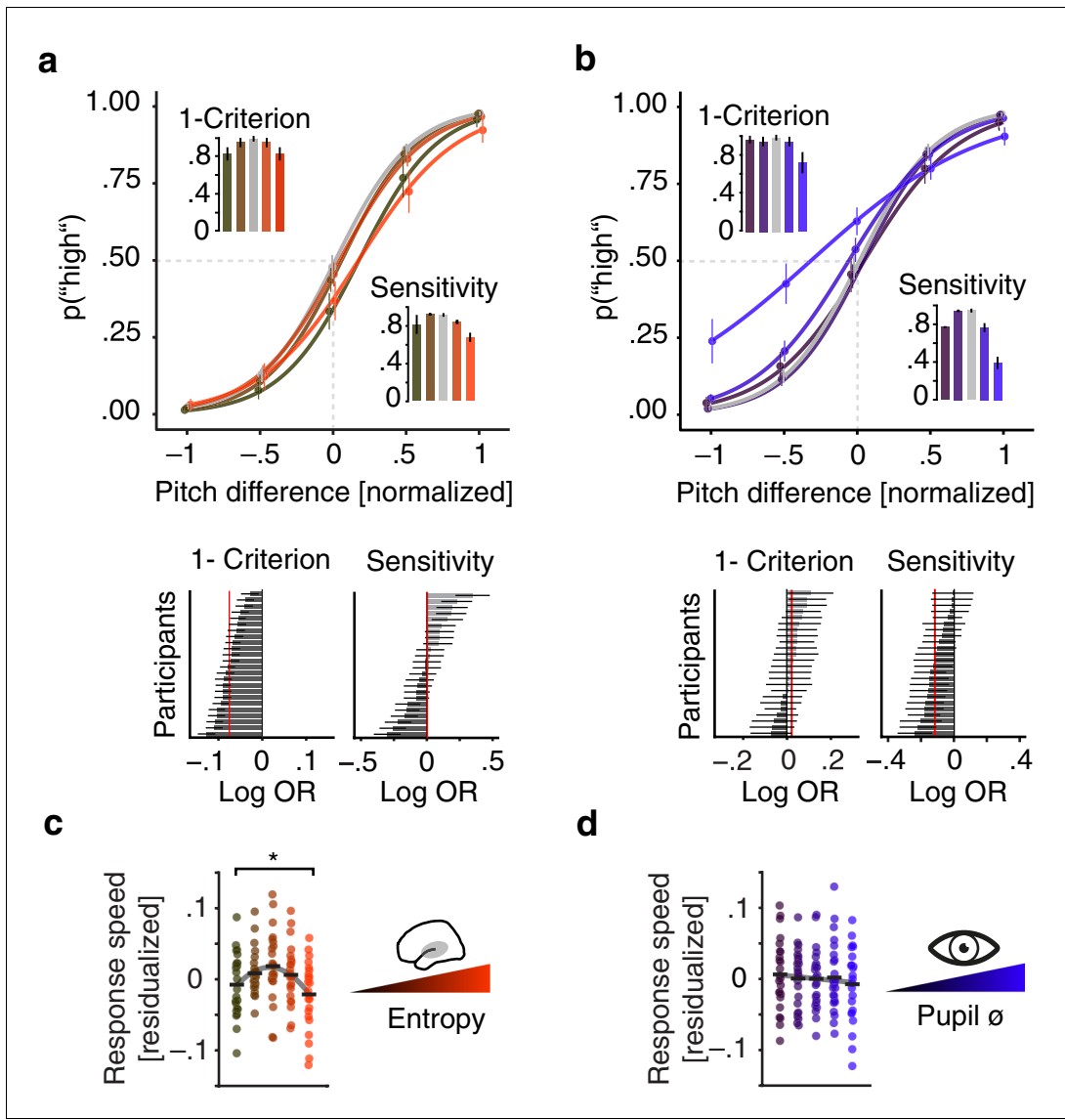

**Figure 5.** Effects of pre-stimulus entropy and pre-stimulus pupil size on perceptual performance. (a) Fixed effects results: probability of judging one tone as 'high' as a function of pitch difference from the median (normalized), resulting in grand average psychometric functions for five bins of increasing entropy (red colours) including point estimates ± 1 SEM. Dashed grey lines indicate bias-free response criterion. Insets show 1–criterion (upper) and sensitivity estimates (lower) ±2 SEMs. Bottom left panel shows single subject log odds (log OR) for the quadratic relationship of pre-stimulus entropy and response criterion (±95 % CI), bottom right panel single subject log ORs for the quadratic relationship of pre-stimulus entropy and sensitivity. Participants sorted for log OR, red line marks fixed effect estimate. (b) As in (a) but for five bins of increasing pre-stimulus pupil size. (c) Single subject (dots) and average response speed (black lines) as a function of increasing pre-stimulus entropy (five bins). (d) As in (c) but as a function of pre-stimulus pupil size. Again, all binning for illustration only. *p<.005.

The online version of this article includes the following figure supplement(s) for figure 5:

**Figure supplement 1.** Overview of fixed and random effects.
**Figure supplement 2.** Single participant psychometric functions.

criterion (log odds (log OR)=−0.06, SE = 0.02, p=0.02; *Figure 5a*, *Supplementary file 10*) and response speed (β = −0.012, SE = 0.004, p=0.002; *Figure 5c*, *Supplementary file 12*) A reduced model that allowed the inclusion of single-subject effects as random slopes revealed that this negative quadratic effect of entropy on response criterion was observable in all participants (see *Figure 5a*). Average predicted response times were lowest following intermediate pre-stimulus entropy (.716 s) compared to low (.762 s) and high (.786 s) entropy. States of intermediate neural desynchronization hence led to a reduction in response time of 50–60 ms compared to high and low desynchronization states.

Conversely, participants proved most sensitive at intermediate levels of arousal: pupil size exhibited negative linear as well as quadratic relations with sensitivity (linear: log OR = −0.232, SE = 0.068, p=0.001; quadratic: log OR = −0.153, SE = −0.035, p<0.001; *Supplementary file 10*) but not with response speed (β = −0.004, SE = 0.003, p=0.1; *Figure 5d*, *Supplementary file 12*). As above, a model including random slopes resulted in negative effects for the vast majority of participants (see *Figure 5b*). Highest sensitivity hence coincided with intermediate arousal and decreased with growing arousal levels.

Like pre-stimulus entropy, pupil size did covary with response criterion. However, the relationship was linearly decreasing (high arousal coincided with a decreased criterion; log OR = −0.115, SE = 0.028, p<0.001; *Figure 5c*) and lacked the marked quadratic relationship observed for pre-stimulus entropy (cf. *Figure 5a*). The increase in bias with arousal was clearly driven by states of particularly high arousal.

In analogy with the approach outlined above for brain–brain models, we computed Wald statistics to assess the distinctness of different quadratic model terms. While response criterion was predicted by EEG entropy following an inverted U shape but not by pupil size ($Z_{Wald}$ = −2.9, p=0.004), response speed was predominantly influenced by pre-stimulus entropy ($Z_{Wald}$ = −1.94, p=0.05). Conversely, pupil size predicted sensitivity better than EEG entropy ($Z_{Wald}$ = 1.6, p=0.1) although this comparison did not yield a statistically significant result. Of note, modelling decisions based on stimulus difficulty alone explained 56.4% of variance (conditional $R^2$) while a model that additionally contained pre-stimulus EEG entropy and pupil size as predictors explained 63.2% of variance in behaviour.

## Control analyses

### Pre-stimulus oscillatory power in auditory cortex does not predict behavioural outcome in the auditory discrimination task

The substantial negative correlation of desynchronization states quantified by entropy on the one hand and low-frequency oscillatory power on the other (see *Figure 3*; *Marguet and Harris, 2011*; *Waschke et al., 2017*) prompted us to repeat the modelling of perceptual performance with pre-stimulus power instead of entropy as a predictor. If entropy only represents the inverse of oscillatory power, effects should remain comparable but change their sign. Oscillatory power however was not significantly linked to behaviour (all p>0.15) and including power as an additional predictor in the model of performance outlined above did not explain additional variance (model comparison; Bayes factor $BF_{Entropy–Power}$ = 98). Thus, local cortical desynchronization but not oscillatory power was linked to perceptual performance.

### Visuo-occipital entropy does not predict behavioural outcome in the auditory discrimination task

To test the cortico-spatial specificity of the outlined desynchronization states to the auditory domain, we repeated all analyses of stimulus-evoked activity and behaviour based on entropy as calculated from visuo-occipital channels. Specifically, we replaced auditory entropy with visual entropy before re-running all relevant models (see Materials and methods for details).

Unsurprisingly, as these spatial filter weights yield imperfect renderings of local cortical activity, we observed a sizable correlation between this visuo-occipital entropy signal and the auditory entropy signal central to our analyses (β = 0.40, SE = 0.009, p<0.001). However, since visual and auditory entropy were also sufficiently distinct (shared variance only $R^2$ = 15%), more detailed analysed on their specific effects were warranted.

We first regressed this visuo-occipital entropy signal on pupil size and observed a weak negative relationship ($\beta = -0.02$, SE = 0.009, p=0.03). Relationships of pre-stimulus entropy over visual cortex with stimulus-evoked auditory activity generally displayed the same direction as for auditory cortex entropy (see *Figure 6a* for summary). Adding to the domain specificity of our main findings, however, visual cortex entropy was a markedly weaker predictor of single-trial phase coherence (model comparison to a model with auditory entropy; Bayes factor $BF_{Auditory-Visual} = 1416$), low-frequency power ($BF_{Auditory-Visual} = 1977$), and gamma power ($BF_{Auditory-Visual} = 39$ see *Figure 6*). Furthermore, visual cortex entropy did not exhibit any relationship with response criterion (log OR = 0.009, SE = 0.02, p=0.66; *Supplementary file 11*). Visual cortex entropy also had no effect on response speed ($\beta = -0.002$, SE = 0.003, p=0.50). Accordingly, auditory cortex entropy explained the response speed data better ($BF_{Auditory-Visual} = 10.8$).

The influence of pre-stimulus desynchronization on stimulus processing and behaviour thus proves to be local in nature, and most selective to desynchronization in sensory regions that are involved in the current task.

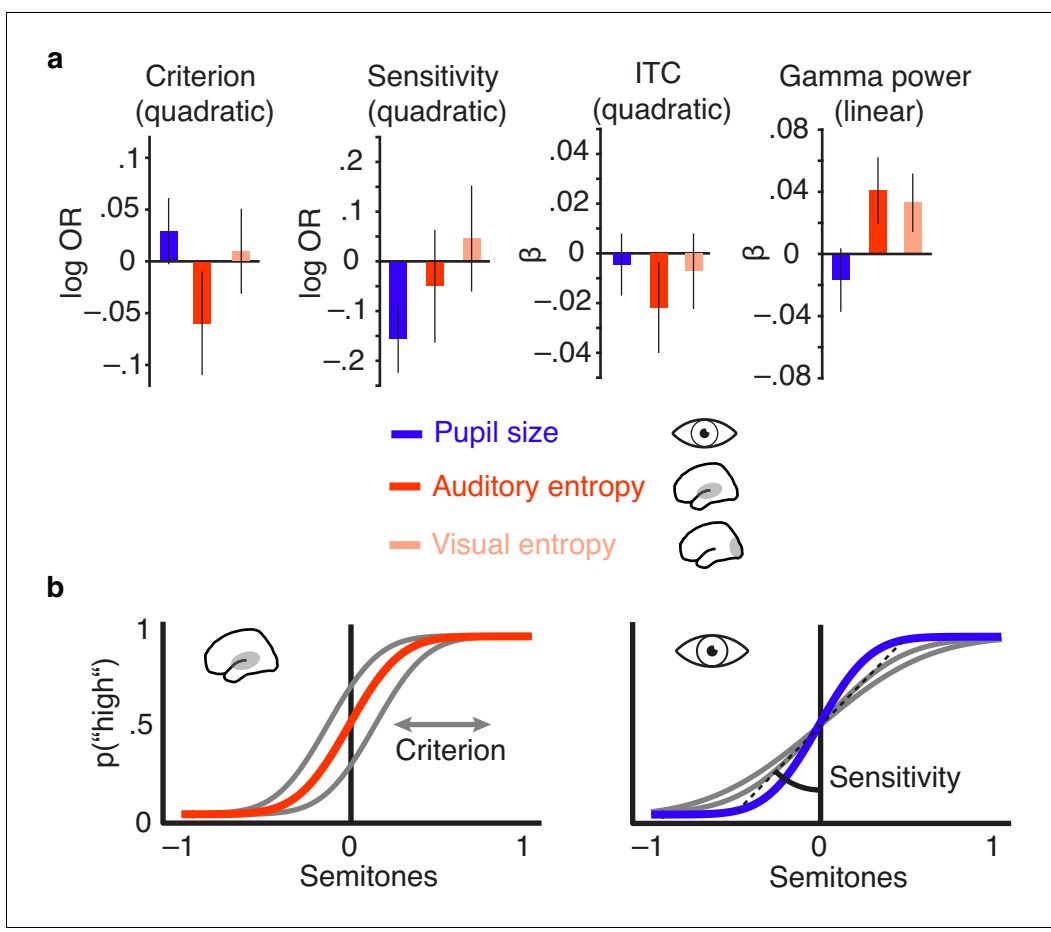

**Figure 6.** Distinct effects of local desynchronization (i.e., auditory entropy) and global arousal (i.e., pupil size). (**a**) Effect sizes (fixed effects, with 95% confidence intervals) for the quadratic relationships of criterion and sensitivity with pupil size (blue), auditory cortex entropy (red) and visual cortex entropy (pale pink). Similarly for the quadratic relationship of pupil size, auditory cortex entropy, and visual cortex entropy with ITC and linear relationships with stimulus-related gamma power. (**b**) Illustrating the quadratic influence of entropy on response criterion (left panel) and pupil size on sensitivity (right panel) by means of an optimal psychometric function (red vs. blue) and non-optimal ones (grey).

The online version of this article includes the following figure supplement(s) for figure 6:

**Figure supplement 1.** Comparison of results from different brain–behaviour models.

## Discussion

This study tested the influence of local cortical desynchronization and pupil-linked arousal on sensory processing and perceptual performance. We recorded EEG and pupillometry, while stimuli of a demanding auditory discrimination task were selectively presented during states of high or low desynchronization in auditory cortex. Desynchronization in auditory cortex and pupil-linked arousal differentially affected ongoing EEG activity and had distinct effects on stimulus-related responses. Furthermore, at the level of single trials, we found unbiased performance and highest response speed to coincide with intermediate levels of pre-stimulus desynchronization and highest sensitivity following intermediate levels of arousal.

### Tracking of auditory cortical desynchronization in real-time

As revealed by the average spatial filter and source projection (*Figure 2*), the signal central to the present analyses mainly originated from auditory cortical areas. The state-detection algorithm we employed was based on entropy of the spatially filtered EEG signal and performed the desired state-dependent presentation with sufficient precision in time (*Figure 2b*). Of note, the distribution used to classify desynchronization states in real-time was updated constantly, which ensured two central prerequisites: First, slow drifts in desynchronization over time were prevented from biasing the state classification. Second, we were able to sample, throughout the experiment, the whole desynchronization state space within each participant (*Jazayeri and Afraz, 2017*). In contrast to an algorithm that sets the criterion for state classification only once per participant and leaves it unchanged thereafter ('open-loop'), the current approach can be referred to as a closed-loop. Technical advances have promoted the use of such closed-loop paradigms to various areas of neuroscientific research, where the main application lies in neurofeedback. Neurofeedback tries to modify behaviour by providing participants with sensory information that is directly proportional to their current brain state (*Sitaram et al., 2017*; *Faller et al., 2019*). Just recently, a number of methodically sophisticated studies have used the power of this approach to relate fluctuations in working memory (*Ezzyat et al., 2018*) or decision making (*Peixoto et al., 2019*) to brain activity in real-time.

### Local cortical desynchronization and arousal differentially shape states of ongoing EEG activity

While there was a pronounced difference in EEG entropy between states of high and low desynchronization, illustrating the power of the used real-time algorithm, no such difference was found for the time-course of pupil size (*Figure 2*). Although pupil size and EEG entropy were positively correlated as has been reported before (*Reimer et al., 2014*), a major part of the variance in EEG entropy was not accounted for by pupil size. We take this as a first piece of evidence that two distinct mechanisms are involved in the generation of perceptually relevant brain states.

The dissociation of both processes is further corroborated by the difference in their respective autocorrelations. Auditory cortical desynchronization displayed a narrower autocorrelation function than pupil size (*Figure 2b*), suggesting two different time scales of operation. Such a finding aligns with a recent study that suggests at least two different time scales that together shape neural activity (*Okun et al., 2019*). On the one hand, fast fluctuations have been suggested to depict synaptic activity and potentially trace back to thalamo- or cortico-cortical interactions (*Haider and McCormick, 2009*; *Harris and Thiele, 2011*). On the other hand, slow fluctuations potentially depict the influence of arousal or neuromodulatory activity in general (*Okun et al., 2019*). While states of local desynchronization likely operate on short time scales in the range of several hundred milliseconds, pupil-linked arousal states rather stretch across several seconds.

Furthermore, changing degrees of desynchronization and arousal manifested in diverse ways in the ongoing EEG: On the one hand, desynchronization in the pre-stimulus time window was negatively related to concurrently measured oscillatory power over a wide range of frequencies (*Figure 3*). The strong negative relationship with low-frequency power replicates previous findings and is tightly linked to the concept of entropy (*Waschke et al., 2017*). On the other hand, pupil-linked arousal in the same time window was negatively linked to low-frequency power, an association frequently observed in invasive recordings of non-human animals (*McGinley et al., 2015b*; *Vinck et al., 2015*). Additionally, arousal was positively related to oscillatory power in the beta band but not in the gamma band. This link of arousal and beta power in EEG differs from reports of a positive

relationship between gamma power of local field potentials (LFP) and pupil size (*Vinck et al., 2015*). Of note, *Vinck et al. (2015)* correlated pupil diameter and LFP gamma power over time within an event-locked time period. In contrast, we related the average pupil diameter in a pre-stimulus time window to spontaneous EEG gamma power across trials. Upon further experimentation, differing methods thus pose the most parsimonious reason for this seeming disparity.

Taken together, the distinct relationships that desynchronization and arousal entertain with key, frequency-domain metrics of instantaneous EEG activity emphasize their independence. We take this as additional evidence for two distinct mechanisms of origin.

## Neurophysiological and neuromodulatory processes of desynchronization and arousal

How plausible is this idea of at least two, at least partially segregate drivers of perceptually relevant brain state? LC–NE activity has been proposed to reflect changes in arousal captured by variations in pupil size (*Aston-Jones and Cohen, 2005*). Although fluctuations in pupil size have recently been linked to activity in the superior colliculus (*Wang et al., 2012*) or the ventral tegmental area (*de Gee et al., 2017*) and also carry information about cholinergic activity (*Reimer et al., 2016*), converging evidence suggests a tight connection to LC–NE activity (*Aston-Jones and Cohen, 2005*; *Joshi et al., 2016*; *Reimer et al., 2016*; *de Gee et al., 2017*). At the same time, in addition to adrenergic and cholinergic projections from brain-stem nuclei, glutamatergic cortico-cortical and thalamo-cortical feedback connections have been proposed as a source of varying states of desynchronization (*Harris and Thiele, 2011*). The widespread NE projections from LC (*Aston-Jones and Cohen, 2005*) are a likely cause for the demonstrable effects of NE-linked arousal on sensory encoding in both the auditory (*McGinley et al., 2015a*) as well as visual domain (*Vinck et al., 2015*). This rationale would thus predict that arousal states should not differ substantially between different sensory cortical regions.

However, modulatory effects of arousal have been found to depend on the experimental context as well as on the sensory modality (*Pakan et al., 2016*; *Shimaoka et al., 2018*). The weak correlation of desynchronization and arousal might thus trace back to our focus on auditory cortical areas. An imperfect direct arousal–desynchronization link in the present data becomes more plausible if we take into account the important distinction between global and local brain states: While the overall level of arousal should have widespread but modality- and context-specific impact on sensory processing and behaviour (*Aston-Jones and Cohen, 2005*; *McGinley et al., 2015b*), the desynchronization of local sensory neural populations could be largely unrelated to, and take place on top of, those global changes (*Beaman et al., 2017*).

Such rather local and modality-specific changes in desynchronization have been assumed to arise from both thalamo- and cortico-cortical feedback connections that represent the allocation of selective attention (*Harris and Thiele, 2011*; *Zagha et al., 2013*; *Zagha and McCormick, 2014*). More precisely, glutamatergic projections between thalamus, prefrontal, and sensory cortical areas might shape the local net degree of inhibition in populations of sensory neurons via AMPA and NMDA receptors and hence influence time-varying local desynchronization. In fact, contingent on the specific task structure, selective attention increases desynchronization in neurons with stimulus-related receptive fields but also across a broader range of task-relevant neurons (*Cohen and Maunsell, 2009*; *Cohen and Maunsell, 2011*). In keeping with this, desynchronization over auditory but not visual cortical areas predicted sensory processing and performance (*Figure 6*). A next step would thus be to combine the present setup for desynchronization–dependent stimulation with manipulations of selective attention. Additionally, future studies might combine single-cell and macroscopic recordings of brain activity with either the monitoring of neurotransmitter release or targeted pharmacological interventions. In the present design we were unable to directly test an involvement of specific neuromodulators in variations of E/I balance and the generation of desynchronization states. Noradrenergic and cholinergic neuromodulation however, have been suggested as a candidate mechanism underlying such dynamics (*Froemke, 2015*).

All things considered, the involvement of two partially related mechanisms in the concomitant generation of desynchronization and arousal states appears likely. On the one hand, desynchronization states presumably are shaped by feedback connections that could result from fluctuations in selective attention (*Harris and Thiele, 2011*). On the other hand, pupil-linked arousal states at least partially hinge on varying levels of LC–NE activity (*Joshi et al., 2016*; *Reimer et al., 2016*) which are

propagated via vast projections towards most regions of cortex and which might be related to over-all changes in the availability of cognitive resources.

If local cortical desynchronization and arousal indeed originate from two distinct processes that both entail functional and behavioural relevance, they should not only have differential effects on the processing of sensory information but also on perceptual performance — which is what we observed here, as discussed next.

## Sensory processing is distinctly affected by desynchronization and arousal states

Desynchronized cortical states have previously been associated in the rodent with enhanced encoding of auditory stimuli (*Marguet and Harris, 2011*), more reliable neural responses (*Pachitariu et al., 2015*), and improved perceptual performance (*Beaman et al., 2017*). Instead, when optogenetically inducing synchronization, perception is impaired (*Nandy et al., 2019*). Conversely, arousal been linked to increased sensory processing of visual stimuli in mice (*Neske and McCormick, 2018*) and humans (*Gelbard-Sagiv et al., 2018*). However, perceptual performance was found to be highest at either intermediate (*McGinley et al., 2015a*; *Neske and McCormick, 2018*) or maximum arousal levels (*Gelbard-Sagiv et al., 2018*).

In the current study, desynchronization and arousal had clearly dissociable effects on sensory processing and behaviour at the single-trial level. First, phase-locked responses were strongest following intermediate levels of pre-stimulus desynchronization (*Figure 4*). Strikingly, this relationship of desynchronization and sensory processing was mimicked by perceptual performance: Intermediate desynchronization led to optimal response criterion and response speed, hence yielding minimally biased and fastest performance (*Figure 5*). Similarly, sensory-evoked gamma power increased with pre-stimulus auditory cortical desynchronization and showed a trend to saturate at intermediate levels. Second, pre-stimulus levels of pupil-linked arousal did only substantially affect sensory-evoked activity in the alpha band but not in low frequencies and were linked to perceptual sensitivity.

Of note, the described tri-fold association of desynchronization, stimulus-evoked activity, and response criterion is generally in accordance with a number of recent studies researching the influence of pre-stimulus oscillatory power on perceptual decisions. Generally, pre-stimulus power in the EEG has been found to bias choice behaviour (*Kayser et al., 2016*). More specifically, however, alpha power (8–12 Hz) prior to stimulus onset has been tightly linked to changes in response criterion and confidence (*Iemi et al., 2017*; *Samaha et al., 2017*; *Wöstmann et al., 2019*). Pre-stimulus alpha power is hypothesized to represent changes in baseline excitability, linking it to response criterion following an inverted u-shaped relationship (*Rajagovindan and Ding, 2011*; *Kloosterman et al., 2019*). These previous findings and the here reported connection of desynchronization and response criterion might at least partially trace back to the same underlying mechanism: that is, task- and attention-specific input to sensory cortical regions via efferent projections leading to a change in net inhibition.

However, only EEG entropy but not oscillatory power was linked to perceptual performance. One reason behind this pattern of results potentially lies in the different contributions both measures receive from time-domain EEG recordings. While alpha power is commonly approximated using a Fourier transform that quantifies the energy of periodic signal fluctuations, EEG entropy receives contributions from periodic as well as aperiodic signal parts. Thus, EEG entropy potentially poses a more sensitive proxy of underlying neural processes than oscillatory power and explains more behavioural variance. Additionally, the task employed in the present study asked participants to integrate sensory evidence presented on a given trial into a reference frame of several tones. This approach differs from commonly used paradigms in the context of pre-stimulus alpha power which typically present stimuli close to the perceptual threshold in simple detection paradigms (e.g., *Iemi et al., 2017*). This difference in experimental tasks could further explain the irrelevance of oscillatory power to behaviour in the present dataset.

Furthermore, although both our present measures of brain state, EEG entropy and pupil size, were positively associated with stimulus-related EEG activity, they affected phase-locked and non-phase-locked brain responses as well as behaviour in distinct ways (see *Figure 6a*). Be reminded, however, that all effects on behaviour and stimulus-related activity were not obtainable when replacing auditory entropy with measures of auditory oscillatory power or with visuo–occipital entropy instead, which underlines their specificity.

Effectively, desynchronization and arousal might interact separately with the two, long-debated building blocks of sensory evoked responses: phase resetting of low-frequency oscillations and additive low-frequency activity (*Shah, 2004*; *Sauseng et al., 2007*). The positive link between phase-locked responses and desynchronization replicates previous findings from our group (*Waschke et al., 2017*) and, combined with the observation of maximum phase coherence following intermediate desynchronization, indicates enhanced early processing of auditory information. Tones presented into states of intermediate desynchronization thus led to a stronger phase-reset.

## Auditory cortical desynchronization and pupil-linked arousal differentially impact performance

Importantly, the dissociation in neural sensory processing parallels a dissociation in behaviour. First, and analogous to the precision of sensory encoding which was highest at intermediate desynchronization levels, responses were least biased following intermediate desynchronization states. This striking parallel in neural and behavioural results cautiously suggests a change in the precision of representations that depends on the current desynchronization state. Second, the impact of arousal on post-stimulus alpha power and perceptual sensitivity, in the light of earlier interpretations (*Voigt et al., 2018*) proposes a similar mechanism: in addition to a clearer early representation of sensory information, intermediate arousal might optimize the integration of such a representation into an existing reference frame. This integration likely involves cortico-cortical feedback connections (*Tallon-Baudry, 1999*) and is essential to allow sensitive perceptual decisions. A different experimental design that allows the direct investigation of the proposed mechanisms represents a crucial next step to understanding the specific functioning of perceptually relevant brain states on the level of sensory neurons.

However, the relationship of arousal and perceptual performance takes a different shape than the respective link to sensory evoked activity might have suggested. While arousal covaried monotonically with post-stimulus activity in the alpha band (and in a statistically non-significant way also in low frequencies, 1–8 Hz), sensitivity was highest at intermediate levels of arousal, testimony to the classic Yerkes–Dodson law. A possible concern might be that we did not sample the state space of pupil-linked arousal in its entirety and hence ended up with a distribution that only captures the lower half of an underlying inverted u (*Faller et al., 2019*), resulting in a positive linear relationship between pupil-linked arousal and post-stimulus low-frequency power. The effect of arousal on sensitivity however did follow an inverted u-shape, suggesting that we indeed sampled a whole range of arousal states.

Additionally, a number of previous observations do in fact match this seeming disarray of stimulus-related activity (increasing monotonically with arousal) and ideal performance (depending quadratically on arousal). First, relatively highest levels of responsiveness in auditory cortical neurons overall can entail the loss of response specificity crucial for precise encoding and perception (*Otazu et al., 2009*). Second, and in line with this rationale, over-amplified responses to auditory stimuli have been linked to age-related decreases of cortical inhibition (*Herrmann et al., 2018*). States of high arousal could thus in principle lead to a similar process of over-amplification and hence prove detrimental to sensory encoding and perception. Third, a recent experiment researching the impact of arousal on visual processing in mice yielded a highly similar pattern of results (*Neske and McCormick, 2018*). *Neske and McCormick (2018)* highlight the role of noradrenergic projections which might transmit task-related activity most efficiently at intermediate arousal levels (*Aston-Jones and Cohen, 2005*).

## Two interrelated systems of local and global brain state jointly shape perception

We here have presented evidence for a joint role of local cortical desynchronization and arousal in the formation of brain states optimal for perceptual performance. The data are commensurate with a model where, on the one hand, arousal shapes global brain states via afferent noradrenergic projections and predominantly influences sensitivity. Conversely, we see local cortical desynchronization in task-related sensory areas to generate local states via attention-dependent feedback connections and to impact response criterion and speed.

To facilitate future research and offer testable hypotheses we intend to leave the reader with some speculations: How could those two mechanisms find an implementation in populations of task-involved sensory neurons? It has been suggested that the shared variability of neuronal populations and its impact on the responses of single neurons are shaped by an additive and a multiplicative source of variation in neural gain (*Arieli et al., 1996*; *Scholvinck et al., 2015*). Whereas a multiplicative gain factor would lead to an overall change in tuning width, an additive factor could create an offset which is believed to differ between neurons (*Lin et al., 2015*). Instantaneous fluctuations of cortical activity, or local cortical desynchronization, are believed to have an additive effect on evoked responses (*Arieli et al., 1996*). Furthermore, arousal-related LC–NE activity exerts a multiplicative influence on the tuning of sensory neurons, which has been suggested to entail relatively sharper tuning curves (*Mather et al., 2016*). However, recent findings challenge this view by showing pupil-linked arousal-related broadening of sensory neural tuning curves (*Lin et al., 2019*). Additionally, it is unlikely that either additive or multiplicative factors alone are the sole source of variability in stimulus-related activity and behaviour (*Lin et al., 2015*). However, the present data allow the testable prediction that selective attention and desynchronization primarily exert an additive influence on neural gain, while LC–NE activity and arousal impact neural gain in a multiplicative fashion.

In sum, the present data provide evidence that, at the single-trial level in humans, desynchronization in sensory cortex (expressed as EEG entropy) and pupil-linked arousal differentially impact sensory and perceptual processes, but jointly optimise sensory processing and performance.

## Materials and methods

### Participants

25 participants (19–31 years, mean age 24.6 years,±3.5 years SD; 10 male) with self-reported normal hearing took part in the experiment. We did not perform a formal power analysis. Importantly, all analyses were based on within-subject effects. Thus, we aimed for a high number of trials per subject (N > 400) to minimize within-subject measurement uncertainty (*Baker et al., 2019*). Participants gave written informed consent and were financially compensated. None of the participants reported a history of neurological or otological disease. The study was approved by the local ethics committee of the University of Lübeck and all experimental procedures were carried out in accordance with the registered protocol.

### Stimulus material

Sets of seven pure tones (±3 steps around 1 kHz; step sizes determined individually, 100 ms duration, 10 ms rise and fall times, sampled at 44.1 kHz) for the main experiment and an additional set of 7 pure tones for the auditory localizer task were created using custom Matlab code (R2017a; MathWorks, Inc, Natick, MA). Initial stimulus frequencies consisted of six steps (±0.27,±0.2, and ±0.14 semitones) around the median frequency (1 kHz) but were adjusted during an individual tracking procedure described below. Stimuli were presented via air conducting in-ear head phones (EAR-TONE 3A), Psychtoolbox and a low latency audio card (RME Audio). All stimuli were presented perfectly audible at a comfortable loudness level approximating 60 dB SPL.

### General procedure

Participants were seated in a quiet room in front of a computer screen. First, they completed an auditory localizer task. Second, participants practiced the main task where, in every trial, they compared one tone against the set of seven tones regarding its pitch and difficulty was adjusted to keep performance at approximately 75% correct. Finally, participants performed 10 blocks of pitch discrimination against an implicit standard (the median pitch, 1 kHz) while tone presentation was triggered by the detection of high or low desynchronization states as outlined below.

### Auditory localizer task

Participants listened to 350 pure tones (six standards, range, 1000–1025 Hz; one oddball at 1050 Hz) separated by inter-stimulus intervals (ISIs) between 1 s and 1.4 s (uniformly distributed). Their task was to detect and count high pitch oddballs (1050 Hz, 50 tones). No overt responses were given during the uninterrupted presentation of tones.

## Main experiment

During each trial, participants were presented with one tone out of the same set of seven pure tones and had to decide whether the presented tone was either high or low in pitch with regard to the whole set of stimuli. In other words, participants implicitly compared each incoming tone to the median frequency in the tone set (i.e., 1000 Hz; *Johnson, 1949*). To hold task difficulty comparable across individuals, up to four rounds of individual tracking (50 trials each) were carried out where the width of the pitch distribution was adjusted depending on performance after each round. Precisely, the width of the pitch distribution was increased (or decreased) if percentage correct was below 70% (or above 80%, respectively). The set of stimuli used during the last round of the tracking procedure was also used during the main experiment.

## Pitch discrimination task

Participants were asked to indicate after each tone whether it was high or low in pitch relative to the whole set of stimuli by pressing one of two buttons of a response box (The Black Box Toolkit). Button orientation was reversed for 13 out of 25 participants. They were instructed to answer as fast and as accurate as possible as soon as the tone had vanished and the response screen had appeared. No feedback was given regarding their performance. A grey fixation cross was presented in the middle of the screen throughout the whole experiment which flickered for one second if participants failed to give a response within 2 s after stimulus offset. Participants performed 60 trials per stimulus levels, resulting in 420 trials split up into 10 blocks of 42 trials each. Every block comprised 6 repetitions of each stimulus level in random order. Note that since the exact time point of stimulus presentation was determined depending on current brain states as identified by the real-time approach outlined below, the average tone-to-tone interval varied between individuals (9.14 ± 1.04 s; min = 8.28 s, max = 12.32 s). Visual presentation and recording of responses was controlled by Psychtoolbox.

## Data recording and streaming

While participants were seated in a dimly lit, sound attenuated booth, EEG signals were measured with a 64-channel active electrode system (actichamp, BrainProducts, Germany). Electrodes were arranged according to the international 10–20 system and impedances were kept below 10 kΩ. Data were sampled at 1 kHz, referenced to electrode TP9 (left mastoid), and recorded using Labrecorder software, part of the Lab Streaming Layer (LSL; *Kothe, 2014*), also used to create a stream of EEG data, accessible in real-time.

Additionally, eye blinks were monitored and pupil size was recorded by tracking participants' right eye at 500 Hz (Eyelink 1000, SR Research). Pupil data was recorded using Eyelink software on a separate machine but at the same time streamed via a TCP/IP connection to the personal computer that was used for EEG recording, brain-state classification, and stimulus presentation. All recorded data was thus available on one machine.

## Spatial filtering and source localization

To focus on EEG activity from auditory cortices, a spatial filter was calculated based on the data from the auditory localizer task of each participant excluding oddball trials. After re-referencing to the average off all channels, we applied singular value decomposition based on the difference between a signal covariance matrix (estimated on EEG data from 0 to 200 ms peristimulus) and a noise covariance matrix (−200–0 ms peristimulus). This approach resulted in a 64 × 64 matrix of eigenvalues and the elements of the first eigenvector were used as filter weights (for similar approaches see *de Cheveigné and Simon, 2008*; *Herrmann et al., 2018*). Matrix multiplication of incoming EEG signals with the spatial filter weights resulted in one virtual EEG channel which largely reflected activity from auditory cortical regions.

To validate this approach, we source localized the same EEG data that was used to construct the signal covariance matrix. To this end, lead fields were computed based on a boundary element method (BEM) template and default electrode locations. Routines from the fieldtrip toolbox (*Oostenveld et al., 2011*) and custom code were used to calculate the sLORETA inverse solution (*Pascual-Marqui, 2002*) which was projected on to the pial surface of a standard structural template (MNI). Arbitrary source strength values were masked at 70% of the maximum.

## Entropy calculation

We computed weighted permutation entropy (WPE) of spatially filtered EEG-signals in a moving window fashion. WPE is an extension to permutation entropy (PE) which was first developed by *Bandt and Pompe (2002)* that considers the amplitude fluctuations of time-series data (*Fadlallah et al., 2013*) and its calculation is outlined below.

In short, WPE approximates the complexity or desynchronization of any neural time-series via three steps: First, recorded samples (here: microvolts) are transformed into symbolic patterns of a predefined length and distance (*equation 1*). Second, the probability of occurrence of those patterns within a snippet of data is used to calculate one entropy value (*Bandt and Pompe, 2002*). Finally, the amplitude information which is lost during the mapping into symbolic space is partially reintroduced by weighing each patterns probability of occurrence by the relative variance of its corresponding neural data (*equations 3* and 4; *Fadlallah et al., 2013*).

In detail, consider the time-series $\{xt\}_{t=1}^{T}$ and a representation incorporating its time delayed sampling $X_j^{m,T} = \{x_j, x_{j+\tau}, \ldots, x_{j+(m-1)\tau}\}$ for $j = 1, 2, \ldots, T - (m-1)\tau$ where $m$ is the so called "motif length" and $\tau$ its "time delay factor". The use of both results in a subdivision of the time series into $N = T - (m-1)\tau$ sub-vectors. Each of those $N$ sub-vectors is mapped into symbolic space by replacing every element with its rank in the respective sub-vector. Note that the total number of possible motifs ($m!$) is limited by the motif length $m$. The probability of occurrence for all possible motifs $\left\{\pi_i^{m,T}\right\}_{i=1}^{m!}$ called $\eth$, which additionally is weighted by $w_j$, can be defined as:

$$p_w\left(\pi_i^{m,\tau}\right) = \frac{\sum_{j \leq N} 1_{u:type(u)=\pi_i}\left(X_j^{m,\tau}\right) \cdot w_j}{\sum_{j \leq N} 1_{u:type(u)=\in\eth}\left(X_j^{m,\tau}\right) \cdot w_j} \tag{1}$$

Note that *type* represents the mapping into symbolic space. Let us furthermore and for simplicity express the weighted occurrence probability of motifs as $P_w = p_w\left(\pi_i^{m,\tau}\right)$. The weighting of probabilities with weight $w_j$ is achieved by calculating the variance of sub-vectors. Therefore we define the arithmetic mean of $X_j^{m,\tau}$ as:

$$\bar{X}_j^{m,\tau} = \frac{1}{m}\sum_{k=1}^{m}\left(X_{j+(K+1)}\tau\right) \tag{2}$$

Each weight value hence is represented by:

$$w_j = \frac{1}{m}\left(X_{j+(k-1)\tau} - \bar{X}_J^{m,\tau}\right)^2 \tag{3}$$

We can finally compute WPE as the Shannon entropy of:

$$H(m,\tau) = -\sum_{i:\pi_i^{m,\tau}\in\eth} P_w \log P_w \tag{4}$$

Since the exact choice of motif length and distance influences the final entropy estimate we relied on recommendations from modelling work and earlier practice (*Riedl and Müller A, 2013*; *Waschke et al., 2017*) by setting the motif length to three and the distance to one (number of samples). To ensure approximation acuity but to retain a high time-resolution, a 200-samples window was moved along the EEG signal in steps of 10 samples, resulting in an entropy sampling rate of 100 Hz.

## Real-time brain-state classification and stimulus triggering

Neural desynchronization in auditory cortical regions was estimated by buffering the EEG signal into Matlab, re-referencing to the average of all channels, applying the individual spatial filter and calculating a time-resolved version of WPE (for details see above).

The resulting entropy time-series was used to generate online a distribution of entropy values. Importantly, this distribution was updated constantly such that it never depended on values older than 30 s. This way, changes in neural desynchronization on longer time-scales were excluded and, instead of a strictly bimodal distribution, the whole desynchronization state space was sampled.

Accordingly, trials with stimuli presented at essentially all levels of absolute desynchronization were obtained (see *Figure 2*). Desynchronization states were defined as a minimum of 10 consecutive entropy samples (100 ms) higher or lower than 90% of the current distribution. Elsewhere in the paper, we will refer to these as high and low states, respectively.

Organized activity in the EEG signal such as evoked responses or eye blinks results in neural synchronization and thus in a drastic reduction in entropy. Although the contribution of eye blinks to the online-analysed EEG signal was minimized by the spatial filter approach, we ensured that no periods containing eye blinks distorted the classification of desynchronization states. To this end, pupil data was read out in real-time and whenever a blink was detected by the eye tracker or pupil size was close to zero, a 'mute' window of 1 s was initiated where incoming EEG data were not considered further. EEG signals immediately following a blink thus were excluded from both, entering the desynchronization distribution and from being classified as a high or low state.

Whenever a high or low state was detected, a new trial started with the presentation of a pure tone after which the response screen was shown and participants gave their response. Note that each tone was presented equally often during high and low states (30 times, yielding 210 trials per state, or 420 trials in total).

## Pre-processing of pupil data

First, the inbuilt detection algorithm was used to locate blinks and saccades before pupil data were aligned with EEG recordings. Second, signal around blinks was interpolated using a cubic spline before low-pass filtering below 20 Hz and down-sampling to 50 Hz. Third, data were split up into trials (−2.5–3 s peristimulus). Finally, single trial time-courses of pupil size were visually inspected and noisy trials (1.3 ± 1.6%) were removed. For visualization purposes, pupil signals were expressed in percentage of the pre-stimulus maximum within a participant (−0.5–0 s peristimulus). Z-scored pupil data was used as a predictor in brain–brain as well as brain–behaviour models. Due to technical difficulties, data from one subject had to be excluded from further analyses.

## EEG offline pre-processing

EEG pre-processing and analyses were carried out using the Fieldtrip and EEGLAB toolboxes (*Delorme and Makeig, 2004*; *Oostenveld et al., 2011*) as well as custom code in Matlab 2017a. First, and as a preparation for independent component analysis (ICA) only, data were re-referenced to the average of all channels, bandpass filtered between 1 and 100 Hz, subsequently down-sampled to 300 Hz, and split up into 2 s long epochs. Rare events like breaks between experimental blocks and noisy channels were excluded based on visual inspection. Second, data were decomposed into independent components using EEGLAB's runica algorithm. Visual inspection of topographies, time-courses, power spectra, and fitted dipoles (dipfit extension) was used to reject artefactual components representing eye blinks, lateral eye movements, heart rate, muscle and electrode noise. Third, raw, un-processed data were loaded, previously detected noisy channels were removed and data were re-referenced to the average of all channels. ICA weights of non-artefactual components were applied to those data before excluded channels were interpolated. Finally, ICA-cleaned data were band-pass filtered between. 5 and 100 Hz using a zero-phase finite impulse response filter and subsequently epoched between −2.5 and 3 s peristimulus. Single trials were visually inspected and rejected in case of excessive noise. On average one channel (±1 channel, M ± SD), 68.9% (±7%) of all components, and 1.4% (±1.6%) of all trials were rejected.

## EEG time–frequency domain analyses

Single trial complex-valued Fourier representations of the data were obtained through the convolution of cleaned and spatially filtered time-courses with frequency adaptive Hann-tapers (four cycles) with a time-resolution of 100 Hz. Power from 1 to 40 Hz (in. 5 Hz steps) and from 40 to 70 Hz (14 exponentially increasing steps) was calculated by squaring the modulus of the Fourier spectrum and was expressed as change in Decibel (dB) relative to average power in the whole trial (−1 to 1.5 s peristimulus).

Additionally, we calculated inter-trial phase coherence (ITC; $0 \leq ITC \geq 1$) and thus divided Fourier representations by their magnitude and averaged across trials before computing the magnitude of the average complex value. Importantly, since, ITC is only defined for a set of multiple trials but not

for single trials, we computed the single-trial measure of jackknife-ITC (jITC; *Richter et al., 2015*; *Wöstmann et al., 2019*). In short, jITC of one trial is defined as the ITC of all trials but the one in question. Note that a trial highly phase-coherent with all others will result in a relatively low value of jITC, reversing the intuitive interpretation of ITC. In the remainder of this paper, we will thus use the term *single-trial phase coherence* when referring to 1–jITC.

## Control analyses

To test the topographical specificity of EEG entropy, we averaged the re-referenced but otherwise raw EEG signal over seven visuo-occipital channels (PO3, PO4, PO7, PO8, POz, O1, O2). Note that this average of a channel selection (all seven visuo-occipital channels receiving equal weight in the average, while other channels effectively received weight 0) is conceptually not different from the way the more sophisticated, pilot–experiment-based auditory spatial filter was calculated. Subsequently, we calculated EEG entropy of this occipital cluster in the exact same way outlined above for auditory cortical areas. The resulting entropy signal was used to repeat all analyses of stimulus-related activity and behaviour. Precisely, mixed models of ITC, stimulus-related power and behaviour were re-run with visuo-cortical entropy. The performance of those models was evaluated by comparing them to the models based on auditory cortical entropy.

## Statistical analyses

### General approach

Trial-wise brain–behaviour and brain–brain relationships were analysed using (generalized) linear mixed-effects models (see below). We used single trial estimates of pre- and post-stimulus brain activity as well as binary decisions ('high' vs. 'low') as dependent variables. Pre-stimulus entropy and pupil size served as predictors. To allow for an illustrative presentation of single subject data, dependent variables were binned based on predictor variables (see *Figure 3*). Note that both EEG signals and behaviour were modelled based on single trial measures of entropy and pupil size, without dichotomizing them into high and low states. Importantly, a contrast between high and low states (for entropy and pupil size) as well as binning was used for visualization only (see *Figures 3* and *4*) and was not part of any statistical analyses reported here. However, single subject fits across bins (of varying number; varying the number of bins between 3 and 7) qualitatively replicated effects of single-trial models.

### Brain–behaviour relationships

As the main interest of this study lay in the influence of pre-stimulus desynchronization and pupil-linked arousal on perceptual sensitivity and response criterion, we combined a generalized linear-mixed-effects model approach with psychophysical modelling: single trial responses (high vs. low) of all participants were modelled as a logistic regression in R (*R Development Core Team, 2018*) using the lme4 package (*Bates et al., 2015*) and a logit link function. The main predictors used in the model were (1) the normalized pitch of presented tones (with respect to the median frequency, seven levels), (2) pre-stimulus entropy (averaged between −0.2 and 0 s peristimulus) and (3) pre-stimulus pupil size (averaged between −0.5 and 0 s peristimulus). Pre-stimulus entropy and pupil size entered the model as both linear and quadratic predictors allowing us to test for non-linear relationships. We additionally included baseline entropy of each trial (3 s pre-stimulus) as a covariate to account for slow fluctuations in average entropy across the duration of the experiment. Note that such an approach is not only in line with current recommendations in statistical literature (*Senn, 2006*) but also comparable to the common inclusion of polynomials in models of functional imaging data (*Kay et al., 2008*). Additionally, a recent study highlighted the superiority of such an approach compared to traditional baseline subtraction in the context of EEG data (*Alday, 2019*). To control for the influence of task duration, trial number was added as a regressor of no interest.

Note that, in the resulting model, a main effect of pitch corresponds to the presence of psychometric response behaviour itself (probability of 'high' responses across pitch levels), a main effect of another predictor (e.g. pupil size) represents a shift in response criterion, and an interaction of pitch and another predictor depicts a change in the slope of the psychometric function, that is a change in sensitivity. Of note, we refrain from interpreting the effects of covariates such as trial number or baseline entropy, as is good practice. For a similar approach and argument see *Alday (2019)*.

Response times were measured relative to the offset of the presented tone and analysed irrespective of response outcome (correct vs. incorrect). To eliminate the impact of outliers, response times below. 2 and above 2 s were excluded from further analyses (*Ratcliff, 1993*). Effects of pre-stimulus desynchronization and arousal on response speed (the inverse of response time, measured in 1/s) were analysed within a linear mixed-effect model framework. Hence, single-trial measures of response speed across all participants were considered as the dependent variable. This analysis approach allowed us to control for a number of other variables including trial number and task ease by adding them as regressors to the model.

## Brain–brain relationships

To test the relationships between neural desynchronization and pupil-linked arousal with ongoing brain activity as well as auditory evoked responses we followed an analogous approach. Namely, different linear mixed-effects models with pre-stimulus entropy and pupil size as predictors were fitted for (i) pre-stimulus low (1–8 Hz) and (i) high (40–70 Hz) frequency power as proxies of ongoing activity. Similarly, different models were fitted for (i) post-stimulus (0–250 ms) single-trial phase coherence (1–8 Hz), as well as (ii) low and (iii) high frequency total power as measures of auditory evoked activity and stimulus processing (see *Figure 2a*). Of note no other covariates than baseline entropy used to model brain–behaviour relationships were included since none explained any additional variance.

## Model fitting

We employed an iterative model fitting procedure, starting with an intercept-only model, to arrive at the best fitting model (*Alavash et al., 2018*; *Tune et al., 2018*).

Fixed effects were added to the model one at a time and the change in model fit was assessed using maximum likelihood estimation. An analogous procedure was adopted for random effects after the best fitting fixed-effect-only model had been determined. We re-coded single trial pitch by first subtracting the median pitch and subsequently dividing by the new maximum, resulting in –1 and 1 for lowest and highest pitch, respectively, and 0 as the midpoint. We z-scored all continuous variables within participants. In the case of binary response behaviour we used generalized linear mixed-effects models with a logit link function. For all other models we employed linear mixed-effects as distributions of dependent variables were not found to be significantly different from a normal distribution (all Shapiro–Wilk $P$ values > 0.1). $P$ values for individual model terms were derived using the Wald z-as-t procedure (*Luke, 2017*).

As measures of effect size we report log odds ratio (log OR) for models of binary response behaviour end regression coefficients β for all other models alongside their respective standard errors (SE). A log OR of 0 indicates no effect for the regressor under consideration. Bayes factors (BF) were calculated for the comparison of two models with an equal number of terms that differed only in one predictor.

To additionally offer an intuitive comparison of predictors' effects on behavior we directly tested some important differences of model estimates using a Wald test. In short, the Wald statistic puts the difference between two estimates from the same model in relation to the standard error of that difference. The resulting test statistic Z (*Bolker et al., 2009*) can be used to test the null hypothesis of no difference between the two estimates in a respective linear model. Z-values above and below ± 1.96, respectively, were considered statistically significant.

To evaluate the performance of the real-time desynchronization detection algorithm described above, we re-calculated entropy (WPE) in the spatially filtered, un-cleaned EEG signal to then compute subject-wise averages of entropy time-courses for high state and low state trials, respectively. A series of paired t-test was used to examine state differences across time. We adjusted p-values to control for the false discovery rate (*Benjamini and Hochberg, 1995*).

## Acknowledgements

Research was supported by the European Research Council (ERC Consolidator grant 646696 to JO) and a GA. Lienert foundation scholarship (LW). Franziska Scharata, Philipp Seidel, and Simon Grosnick helped acquire the data. We thank Hong-Viet V Ngo for assistance with illustrations, Björn Herrmann for help with source projection, and Santiago Jaramillo for insightful comments on an earlier

version of this manuscript. We additionally would like to thank Jan Willem de Gee, Jonathan Peelle, and an anonymous reviewer for constructive feedback.

## Additional information

### Funding

| Funder | Grant reference number | Author |
|---|---|---|
| H2020 European Research Council | 646696 | Jonas Obleser |

The funders had no role in study design, data collection and interpretation, or the decision to submit the work for publication.

### Author contributions

Leonhard Waschke, Conceptualization, Data curation, Software, Formal analysis, Visualization, Writing - review and editing; Sarah Tune, Methodology, Writing - review and editing; Jonas Obleser, Conceptualization, Resources, Supervision, Funding acquisition, Methodology, Project administration, Writing - review and editing

### Author ORCIDs

Leonhard Waschke https://orcid.org/0000-0002-1248-9259
Sarah Tune http://orcid.org/0000-0001-9022-9965
Jonas Obleser https://orcid.org/0000-0002-7619-0459

### Ethics

Human subjects: Participants gave written informed consent to participate and consent to publish the recorded data in anonymised form. They were financially compensated. The study was approved by the local ethics committee of the University of Lübeck (reference number 15-313) and all experimental procedures were carried out in accordance with the registered protocol.

### Decision letter and Author response

Decision letter https://doi.org/10.7554/eLife.51501.sa1
Author response https://doi.org/10.7554/eLife.51501.sa2

## Additional files

### Supplementary files

• Supplementary file 1. Estimates and statistics of the model predicting pre-stimulus low-frequency power. The table shows model coefficients, standard errors, effect size estimates as well as goodness of fit statistics for the model reported in results and discussion sections.

• Supplementary file 2. Estimates and statistics of the model predicting pre-stimulus alpha power. The table shows model coefficients, standard errors, effect size estimates as well as goodness of fit statistics for the model reported in results and discussion sections.

• Supplementary file 3. Estimates and statistics of the model predicting pre-stimulus beta power. The table shows model coefficients, standard errors, effect size estimates as well as goodness of fit statistics for the model reported in results and discussion sections.

• Supplementary file 4. Estimates and statistics of the model predicting pre-stimulus gamma power. The table shows model coefficients, standard errors, effect size estimates as well as goodness of fit statistics for the model reported in results and discussion sections.

• Supplementary file 5. Estimates and statistics of the model predicting post-stimulus low-frequency power. The table shows model coefficients, standard errors, effect size estimates as well as goodness of fit statistics for the model reported in results and discussion sections.

- Supplementary file 6. Estimates and statistics of the model predicting post-stimulus alpha power. The table shows model coefficients, standard errors, effect size estimates as well as goodness of fit statistics for the model reported in results and discussion sections.

- Supplementary file 7. Estimates and statistics of the model predicting post-stimulus beta power. The table shows model coefficients, standard errors, effect size estimates as well as goodness of fit statistics for the model reported in results and discussion sections.

- Supplementary file 8. Estimates and statistics of the model predicting post-stimulus gamma power. The table shows model coefficients, standard errors, effect size estimates as well as goodness of fit statistics for the model reported in results and discussion sections.

- Supplementary file 9. Estimates and statistics of the model predicting post-stimulus low-frequency phase coherence. The table shows model coefficients, standard errors, effect size estimates as well as goodness of fit statistics for the model reported in results and discussion sections.

- Supplementary file 10. Estimates and statistics of the model predicting decisions (high vs. low). The table shows model coefficients, standard errors, effect size estimates as well as goodness of fit statistics for the model reported in results and discussion sections.

- Supplementary file 11. Estimates and statistics of the model predicting decisions based on visual cortex entropy. The table shows model coefficients, standard errors, effect size estimates as well as goodness of fit statistics for the model reported in results and discussion sections.

- Supplementary file 12. Estimates and statistics of the model predicting response speed. The table shows model coefficients, standard errors, effect size estimates as well as goodness of fit statistics for the model reported in results and discussion sections.

- Transparent reporting form

### Data availability

EEG data and pupillometry data are publicly available on the Open Science Framework (OSF) https://osf.io/f9kzs/. Custom computer code to reproduce all essential findings is publicly available on the OSF https://osf.io/f9kzs/.

The following dataset was generated:

| Author(s) | Year | Dataset title | Dataset URL | Database and Identifier |
|---|---|---|---|---|
| Leonhard Waschke | 2019 | RealNoi | https://osf.io/f9kzs/ | Open Science Framework, f9kzs |

The following previously published dataset was used:

| Author(s) | Year | Dataset title | Dataset URL | Database and Identifier |
|---|---|---|---|---|
| Massimini M, Laureys S | 2017 | Rest EEG recordings in healthy subjects during wakefulness, sleep and anesthesia with ketamine, propofol, and xenon | https://doi.org/10.5281/zenodo.806176 | Zenodo, 10.5281/zenodo.806176 |

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
