## [Decision Letter]

**Acceptance summary:**

During perception, multiple brain systems affect how sensory input is processed. Here the authors concurrently recorded pupil size and electrical brain data to study the interrelation of neural dynamics and noradrenergically-modulated arousal in the context of perceptual decision making. The authors used a closed-loop system to present an auditory decision task during states of high or low neural synchronization in auditory cortex. Neural desynchronization and pupil-measured arousal were differentially related to auditory perceptual processing, suggesting joint influences of global arousal and neural desynchronization on behavior. These findings are therefore important not only for understanding auditory perception, but for our broader understanding of linked neural systems supporting behavior on multiple timescales.

**Decision letter after peer review:**

Thank you for submitting your article "Neural desynchronization and arousal differentially shape states for optimal sensory performance" for consideration by *eLife*. Your article has been reviewed by two peer reviewers, and the evaluation has been overseen by a Reviewing Editor and Laura Colgin as the Senior Editor. The following individual involved in review of your submission has agreed to reveal their identity: Jan Willem de Gee (Reviewer #1).

The reviewers have discussed the reviews with one another and the Reviewing Editor has drafted this decision to help you prepare a revised submission.

Summary:

Waschke and colleagues concurrently record pupil size and electrical brain data to study the interrelation of neural dynamics and noradrenergically modulated arousal in the context of perceptual decision making. The authors used a closed-loop system to present an auditory decision task during states of high or low neural synchronization in auditory cortex. EEG desynchronization and pupil-measured arousal were differentially related to auditory perceptual processing, suggesting joint influences of global arousal and neural desynchronization.

Essential revisions:

1) In Figure 1B it can be observed that neural desynchronization fluctuates much faster than arousal. It would be important to characterize this, for example by computing the autocorrelation of both signals, both at the timeseries and trial-wise scalar levels. If indeed the autocorrelation function for arousal is significantly broader than that for neural desynchronization (as Figure 1B suggests), this may indicate that these measures track different brain states. Overall, a more detailed discussion about the different timescales of measurement and signal processing for the pupillometry and EEG data would be useful.

2) To further characterize the relationship between pre-trial neural desynchronization and pupil-linked arousal by searching the complete time-frequency-electrode space. A large number of studies showed that LC activity (and pupil dilation) is/are linked to low frequency desynchronization and an increase in high frequency power (Marzo et al., 2014; McGinley et al., 2015a; Neves et al., 2018; Stitt et al., 2018), conflicting with the here observed findings (Figure 3 and 4). We would like to see a control analysis which searches for an association between pupil dilation and de/synchronization over the complete time-frequency-electrode space (i.e., not restricted to the (for pupil analyses quite short) pre-stimulus time window and auditory electrodes).

3) Although the 1-8 Hz and 40-70 Hz frequency bins have some theoretical motivation, the absence of alpha and beta results caused some questions about the specificity of what is reported – please include these as well for all main analyses (in supplementary material would be fine).

4) Statistics: It would be useful to show whether quadratic models explain significantly more variance than linear models. And a global comment: Sometimes models are compared reporting p-values (subsection “Pre-stimulus oscillatory power in auditory cortex does not predict behavioural outcome in the auditory discrimination task”), using likelihood ratio tests and sometimes using Bayes factors (subsection “Visuo-occipital entropy does not predict behavioural outcome in the auditory discrimination task”, third paragraph). Please be consistent in your approach (e.g., report the Bayes factor instead of the "all P >.15").

5) Based on the outcome of the above analyses, and based on the additional points reviewers raised regarding the presentation/interpretation of the results (and the way these are currently embedded in the existing literature), portions of the paper will have to be rewritten to accommodate updated results.

---

## [Author Response]

Essential revisions:1) In Figure 1B it can be observed that neural desynchronization fluctuates much faster than arousal. It would be important to characterize this, for example by computing the autocorrelation of both signals, both at the timeseries and trial-wise scalar levels. If indeed the autocorrelation function for arousal is significantly broader than that for neural desynchronization (as Figure 1B suggests), this may indicate that these measures track different brain states. Overall, a more detailed discussion about the different timescales of measurement and signal processing for the pupillometry and EEG data would be useful.

This is a very thoughtful comment. We extended Figure 2 (see panel B) with autocorrelation functions of neural desynchronization (EEG entropy) and arousal (pupil size). In addition to the average autocorrelation of single trial time-courses, we computed across trial autocorrelations of the respective average pre-stimulus values. As is evident from these figures, arousal indeed shows a broader autocorrelation than neural desynchronization. We interpret this as additional evidence for both measures tracking distinct brain states at different time scales. In the updated version of the manuscript the two timescales of neural desynchronization and arousal are additionally discussed in Results and Discussion section. The corresponding text passages are pasted below for convenience.

“Furthermore, auditory cortical desynchronization and pupil linked arousal, as approximated by EEG entropy and pupil size, displayed different autocorrelation functions (Figure 2B). While EEG entropy states were self-similar on an approximate ~500 ms scale, states of pupil size extended over several seconds.”

“The dissociation of both processes is further corroborated by the difference in their respective autocorrelations. […] While states of local desynchronization likely operate on short time scales in the range of several hundred milliseconds, pupil-linked arousal states rather stretch across several seconds.”

2) To further characterize the relationship between pre-trial neural desynchronization and pupil-linked arousal by searching the complete time-frequency-electrode space. A large number of studies showed that LC activity (and pupil dilation) is/are linked to low frequency desynchronization and an increase in high frequency power (Marzo et al., 2014; McGinley et al., 2015; Neves et al., 2018; Stitt et al., 2018), conflicting with the here observed findings (Figure 3 and 4).

This is an important concern and we agree with the reviewers that the absence of the described relationship between pupil-linked arousal and oscillatory remained somewhat puzzling.

One reason for the absence of this effect in the original version of the manuscript potentially lay in our choice of baseline correction parameters for oscillatory power. We had used average oscillatory power in a 500 ms long pre-stimulus window as the baseline, a standard procedure in electrophysiological research. However, subtracting the average pre-stimulus activity might have obscured effects of interest in that very time-window. To investigate such a potential confound, we repeated all analyses using oscillatory power averaged across the whole trial period (–1 to 1.5 s) as the baseline. Note that this approach can come close to using no baseline at all but still allows to express oscillatory power in a meaningful unit (dB).

Corroborating the reviewers’ notion, when a whole-trial baseline is used (as in this revised version), pre-stimulus pupil size displays a negative link with low-frequency power in the pre-stimulus time-window. Furthermore, and prompted by the additional analyses requested in major query 3, pre-stimulus pupil size is positively linked to pre-stimulus beta power.

All other results of models predicting ongoing and stimulus-evoked activity remained qualitatively unchanged. We updated all figures, tables and statistics reported in the main text and supplements accordingly.

The results presented in the updated version of the manuscript are in line with previous studies that report a negative association between pupil-linked arousal and the power of low-frequency oscillations. Although our finding of a positive link between pupil size and beta but not gamma power does not align perfectly with earlier results, we deem it possible that the difficulties of recording high quality signals at high frequencies in the EEG might represent one reason for the absence of a positive link between pupil size and broad band high frequency power.

We would like to see a control analysis which searches for an association between pupil dilation and de/synchronization over the complete time-frequency-electrode space (i.e., not restricted to the (for pupil analyses quite short) pre-stimulus time window and auditory electrodes).

To follow up on the reviewers’ question regarding a time-frequency-electrode resolved analysis of the relationship between oscillatory power and pupil size, we chose the following approach:

First, at the level of single subjects, we performed a multiple regression with single trial oscillatory power as dependent variable and pre-stimulus entropy, baseline entropy and pre-stimulus pupil size as predictors. Notably, one regression was calculated for each time, frequency, and channel bin. At the second level, the resulting regression coefficients of pupil size were tested against zero using a cluster-based permutation approach (Maris and Oostenveld, 2007).

Indeed, one negative cluster that was limited to low frequencies but extended over a wide pre-stimulus time-window emerged (see Author response image 1). In convergence with our results outlined above, the topography of this effect included central electrodes but was strongest at occipital sites. Hence, pre-stimulus pupil size was negatively linked with low-frequency power and this relationship was strongest over but not limited to visual cortical areas.

Notably, the pattern of results did not change qualitatively when average pupil size in earlier time-windows was use as a regressor (e.g., -1 to -.5 s). This speaks to the extended duration of pupil-linked arousal states suggested by the autocorrelation functions of pupil size.

**Author response image 1. respfig1:** Effects of pre-stimulus pupil size on oscillatory power. Time-frequency resolved t-values from a cluster test on regression coefficients expressing the relationship between pre-stimulus pupil size (averaged between –.5 and 0 s) oscillatory power (left). Note the negative cluster covering low frequencies (white outlines). T-values averaged over time and frequency points within the significant cluster display a wide-spread topography that peaks over occipito-parietal regions (right).

3) Although the 1-8 Hz and 40-70 Hz frequency bins have some theoretical motivation, the absence of alpha and beta results caused some questions about the specificity of what is reported – please include these as well for all main analyses (in supplementary material would be fine).

We agree with the reviewers that despite the alignment of the used frequency bins with previous research, results for alpha and beta bands potentially are of interest to the reader. Thus, we additionally tested the relationship of pre-stimulus EEG entropy and pupil size with alpha (8–12 Hz) and beta power (14–30 Hz) both for pre- and post-stimulus time-windows. Power was modelled analogously to the approach used for low and high frequency power (cf. Figures 3 and 4).

As can be discerned from Figure 4—figure supplement 1, both pre-stimulus alpha and beta power displayed a negative relationship with pre-stimulus EEG entropy (alpha power: β = -.291, SE = .01, *P* < .001; beta power: β = -.316, SE = .01, *P* < .001). Pupil size on the other hand, while not significantly linked to alpha power (β = .009, SE = .01, *P* = .35) showed a positive relationship with beta power (β = .04, SE = .01, *P* < .001).

These results demonstrate a broad negative relationship of EEG entropy and oscillatory power. Although the strength of this link varies over frequencies, pre-stimulus power overall is negatively related to EEG entropy.

On the other hand, sensory-evoked power in the alpha band was positively linked to pre-stimulus pupil size but not EEG entropy (see Figure 4—figure supplement 2 in the updated version of the manuscript). In line with the trend-level positive relationship of pre-stimulus pupil size and post-stimulus low-frequency power, this suggests that pupil-linked arousal affects the amplitude of sensory-evoked responses. In contrast, auditory cortical desynchronization as measured by EEG entropy increases low-frequency phase-locked responses (see Figure 4) but is negatively related to their amplitude. Hence, these findings are in support of the dissociable influences neural desynchronization and arousal exert on early sensory processing suggested in the original version of the manuscript. While neural desynchronization affects low-frequency phase-coherence, pupil-linked arousal influences post-stimulus low-frequency power.

Finally, sensory-evoked power in the beta band was neither substantially related to EEG entropy nor to pupil size. Thus, the desynchronization-dependent gain of post-stimulus high-frequency power reported in the original version of the manuscript is specific to frequencies above 40 Hz.

4) Statistics: It would be useful to show whether quadratic models explain significantly more variance than linear models.

This is an important issue, and we would like to use this opportunity to outline and motivate our general approach. Although we acknowledge the theoretical relevance of a comparison between linear and quadratic models, our approach did not include a direct comparison of both due to several reasons. However, note that a model that includes both terms (as did essentially all our models) takes an unbiased approach statistically nevertheless, and in fact does allow for a direct comparison, as we will elaborate on below.

First, it is unclear based on previous research whether the interplay between desynchronization and behaviour is best characterized by a linear or a quadratic relationship. Similar things are true for the impact of arousal on performance. In order to test both shapes of relationships in every model, we always included linear as well as quadratic terms based on previous research.

Second, in order to directly compare purely linear and purely quadratic models, one would have to assume a complete dissociation of linear and quadratic terms. In other words, one would assume that a quadratic trend describes a perfectly symmetric parabola or that the linear term in a second-degree polynomial is zero. To avoid such strong and unrealistic assumptions, we always included linear and quadratic terms. Hence, data were fit by second degree polynomials without fixing any coefficients a priori. Note that in such a model linear terms do not change depending on quadratic terms.

Third, we aimed for consistency rather than data-driven variations. Instead of separately deciding for each brain-brain model whether to include a quadratic term or not based on variance explained, we decided to include it in all models in order to keep them comparable. Since significant quadratic effects were apparent for several dependent variables (e.g. phase coherence, response speed or sensitivity) we also modelled them for all others.

In this context we would like to point out that in a model that includes both a linear and a quadratic term, a significant quadratic term is equivalent to the quadratic effect explaining variance above and beyond the linear one. In contrast, this is not the case in a model with a non-significant quadratic term.

Thus, in the light of previous research that suggested linear as well as quadratic influences of neural desynchronization or arousal on perceptual performance we included linear and quadratic trends in all models. Removing them in the case of non-significant terms and reporting the purely linear model instead would, in our opinion, complicate things for the reader.

The way we report model results in the revised version of the manuscript makes it straightforward to assess the impact of linear and quadratic terms, respectively. In sum, the inclusion of quadratic terms in all models is not only essential in order to test previously suggested hypotheses but also entails a high degree of transparency in the reporting of results.

And a global comment: Sometimes models are compared reporting p-values (subsection “Pre-stimulus oscillatory power in auditory cortex does not predict behavioural outcome in the auditory discrimination task”), using likelihood ratio tests and sometimes using Bayes factors (subsection “Visuo-occipital entropy does not predict behavioural outcome in the auditory discrimination task”, third paragraph). Please be consistent in your approach (e.g., report the Bayes factor instead of the "all P >.15").

We apologize for this inconsistency. We report Bayes factors for all model comparisons in the updated version of the manuscript. Nevertheless, we want to highlight that we use P-values only in order to express the significance of model terms such as the effects of oscillatory power on behaviour referred to by the reviewers. Of note, these P-values derive from a Satterthwaite approximation and not from a model comparison based on log-likelihood tests. For transparency, we of course also report the estimated β values for linear models and Odds ratios (OR) for generalized linear models alongside their standard errors.

Although one could want to aim for even more consistency by reporting P-values for both, model comparisons and model terms, this endeavour is complicated by the requirements of log-likelihood tests. In order to compare two models using a log likelihood test, models must be nested and thus must differ in complexity by at least one term, resulting in one degree of freedom. Since this is not the case for most model comparisons performed as part of the current manuscript, we settled on Bayes factors as a stable, transitive, and interpretable metric.

5) Based on the outcome of the above analyses, and based on the additional points reviewers raised regarding the presentation/interpretation of the results (and the way these are currently embedded in the existing literature), portions of the paper will have to be rewritten to accommodate updated results.

We have re-written parts of the Introduction, Results and Discussion in order to account for the reviewers’ suggestion and updated results. Additionally, all figures but Figure 1 were updated. To account for recent scientific findings, we have furthermore re-written parts of the Discussion section. Throughout the updated version of the manuscript, changes are marked in orange.